

# The Role of Atmospheric Rivers in compound events consisting of heavy precipitation and high storm surges along the Dutch coast

Nina Ridder[1], Hylke de Vries[1], and Sybren Drijfhout[1]

[1]Royal Dutch Meteorological Institute, Postbus 201, 3730 AE De Bilt, The Netherlands

*Correspondence to:* ridder@knmi.nl

**Abstract.** Atmospheric river (AR) systems play a significant role in the simultaneous occurrence of high coastal water levels and heavy precipitation in the Netherlands. Based on observed precipitation values (E-OBS) and the output of a numerical storm surge model (WAQUA/DSCMv5) forced with ERA-Interim sea level pressure and wind fields, we find that the majority of compound events between 1979 -2015 has been accompanied by the presence of an AR over the Netherlands. By isolating and assessing the prevailing sea level pressure (SLP) and sea surface temperature (SST) conditions up to seven days before the events with and without AR involvement, we show that the presence of ARs constitutes a specific type of forcing conditions that (i) resemble the SLP anomaly patterns during the positive phase of the North Atlantic Oscillation (NAO+) with a North-South pressure dipole over the North Atlantic and (ii) cause a warming of the western boundary of the North Atlantic. These conditions are clearly distinguishable from those conditions during compound events without the influence of local ARs which occur under SLP conditions resembling the East Atlantic (EA) pattern with a West-East pressure dipole over Northern Europe and are accompanied by a cooling of the West Atlantic. Thus, this study provides a useful tool for the early identification of possible harmful meteorological conditions over the Netherlands and supports effort for the establishment of an early warning system.

## 1 Introduction

Currently, policy decisions to respond to flood risk and its increase under global climate change are based on the assumption that coastal flooding is caused by a single, isolated and independent hazard, e.g. heavy precipitation or high river discharge. However, it has become increasingly obvious that this "single-hazard-approach" is insufficient to account for some of the most extreme flooding events observed over the past decades which were in fact often induced by the combined effect of multiple hazards (e.g. Kew et al., 2013; van den Hurk et al., 2015; Vorogushyn et al., under rev.; Zscheischler et al., 2018). These so called "compound events" generally have a more devastating impact than their single-hazard equivalent and exert significant influence on the relevant flood statistics (van den Hurk et al., 2015; Zscheischler et al., 2018). Understanding the underlying dynamics of compound events is therefore paramount to support policymakers to take informed decisions and implement effective coastal protection measures.

In this study we focus on compound events (CEs) in the form of heavy local precipitation and high surge levels (hereafter simply referred to as CEs) along the Dutch coast. For low lying countries like the Netherlands with a long coastline, understanding CEs



related to coastal flooding is of particular importance as these have the potential to cause catastrophic impacts. First assessments of this type of compound events have aimed their attention mostly to the impact of compound events on flood risk in terms of return period (e.g. Kew et al., 2013; van den Hurk et al., 2015). While all these studies conclude that the exclusion of CEs leads to a sever underestimation of flood risk along the Dutch coast, which renders the application of current assessments for design

standards insufficient, little detail is known about the mechanisms driving the simultaneous occurrence of heavy precipitation and high surge levels. A solid understanding of these processes and their interaction is, however, crucial to understand the implications that future climate change may have on the occurrence of CEs and thus the future development of future flood risk. To close this gap, this study focuses on the large-scale climatologic conditions leading to the simultaneous occurrence of heavy precipitation and high surge levels. In particular, the study aims to identify the importance of one atmospheric phenomenon

that has been suggested to potentially be involved in coastal CEs due to its association with high precipitation and strong near-surface winds, namely atmospheric rivers (Waliser and Guan, 2017).

Atmospheric rivers (ARs) are long filaments of high water vapour concentration typically located in the lower troposphere which travel from low to midlatitudes towards the poles in both hemispheres. They play an important role in the hydrological cycle being responsible for over 90% of the poleward water vapour transport at midlatitudes (Zhu and Newell, 1998; Gimeno

et al., 2014; Guan and Waliser, 2015; Dacre et al., 2015). They develop in relation with extra-tropical cyclones and move with the large-scale dynamic phenomena that produce them (hereafter AR system). The vast geometric extent of ARs with a typical width of 400 - 600 km and lengths of over 2,000 km allows them to cover and affect large geographical areas simultaneously. If these water vapour-rich structures make landfall, orographic lifting (Lavers and Villarini, 2013) and, to a minor extent, other synoptic-scale and mesoscale processes (Ralph and Dettinger, 2012) can cause severe precipitation events that have been

linked to major floods in many geographical regions (e.g. Gimeno et al., 2014, and references therein). In Western Europe landfalling ARs dominate the high tail of extreme precipitation and their impacts can reach as far inland as Poland (Lavers and Villarini, 2013; Waliser and Guan, 2017). The strong near-surface winds associated with ARs constitute up to half of the events in the highest 98th percentile of the wind distribution along the Western European coastline between 1997 and 2014 (Waliser and Guan, 2017). Thus, AR systems have the potential to play an important role in coastal surge heights a characteristic not

previously assessed (Waliser and Guan, 2017).

The determination of the importance of ARs for and their impact on the conditions during coastal CEs in the Netherlands will pave the way to better understand the underlying risk CEs pose for coastal areas and to a possible early identification of hazardous conditions. This is particularly important in the light of the projected frequency enhancement and intensification of ARs under global climate change (Ramos et al., 2016). Despite their importance for flood risk however, even univariate

assessments of the impact of ARs and AR-carrying systems in the Netherlands have been incomplete by focusing on the impact of ARs on local precipitation. While these studies have brought valuable insights into the impact of ARs on precipitation in the Netherlands, there have been no equivalent assessments for the impact of ARs on coastal water level extremes or the connection between water level and precipitation extremes. Thus, it is unclear if the strong winds accompanying ARs can induce storm surges along the Dutch coast, where north-northwesterly winds cause the highest storm surges (Kew et al., 2013, e.g.). This

puts a constraint on the AR-causing low-pressure systems passing over the Netherlands that is not necessarily met by every



one of those.

The study presented here connects the impact of ARs on both, precipitation and coastal surge levels. To achieve this we apply the "bottom-up" approach introduced by Hazeleger et al. (2015), which uses the impact, here the co-occurrence of high water levels and heavy precipitation, as venture point for the analysis and identifies the physical processes driving the particular

impact from there. This approach is particularly suited for compound events as it allows the identification of drivers with the largest impacts (Zscheischler et al., 2018). In detail, we investigate coastal water levels derived from a numerical surge model (WAQUA/DCSMv5) driven by reanalysis data and link these to observed precipitation (EOBS) over the Netherlands from 1979 to 2015. From this dataset we identify CEs by isolating those events where both, precipitation and water level, exceed a pre-defined quantile threshold. In a second step we identify days with the presence of an AR over the Netherlands. We then

compare mean conditions during CEs with and without the involvement of ARs and identify the driving mechanisms behind these two types of CEs. Finally we determine the difference between conditions during ARs associated with CEs and those that are not. In this way, our study provides a first classification for compound events and presents a detailed assessment of conditions leading to coastal CEs in the Netherlands while focusing on the influence of ARs on their driving mechanisms. This will determine the potential of ARs to aggravate hazards related to coastal CEs in the Netherlands and deliver valuable insight

into the atmospheric processes driving these events. The findings of this study could then be used to develop an early warning system using ARs as an indicator for upcoming events.

## 2   Study area

This study focuses on the possibility and significance of ARs systems causing compound events along the Dutch coast. Located largely at or below sea level, the Netherlands does not show any significant orographic features (Fig. 1). As a result, ARs passing

over the Netherlands do not necessarily cause extreme precipitation (Beukema, 2014). Nevertheless, it has been shown that those ARs making landfall along the Dutch coast can lead to significant precipitation events (Waliser and Guan, 2017).

For the impact assessment, our analysis focuses on a selection of four stations spread along the Dutch coast, namely Hoek van Holland (HvH), Den Helder (DHR), Harlingen (HRL) and Delfzijl (DLZ). The catchment areas associated with these stations are shown in Fig. 1 and include the south of the Netherlands (hereafter SNL) for Hoek van Holland, the Lake IJssel and its

surrounding region (hereafter LIJ) for Den Helder and Harlingen, and the north-east of the Netherlands (hereafter NENL) for Delfzijl. The stations were chosen due to their importance in the Dutch water management system and thus, their significance for flood risk in the Netherlands. Further, they represent a spread of stations along the Dutch coast and cover all its orientations. In this way, our study accounts for stations situated at the westward facing part of the coast (HvH), the northward facing part in the Wadden Sea (HRL, DLZ) and one station facing both directions located at the far west corner of the Dutch mainland

(DHR).



## 3  Data and method

### 3.1  ERA-Interim reanalysis dataset

The ERA-Interim reanalysis dataset is produced by the European Centre for Medium-Range Weather Forecast (ECMWF). It is the result of reanalysis simulations performed using a three-component forecast model (Integrated Forecasting System IFS release Cy31r2) for the time period from 1 Jan 1979 to present day (Berrisford et al., 2011; Dee et al., 2011). The IFS uses the spectral grid T255 ($\sim$80 km) and has 60 vertical levels spanning from the surface up to 0.1 hPa. Analysis time steps are provided every six hours for most atmospheric variables, i.e. each day contains information about atmospheric conditions at 00:00, 06:00, 12:00 and 18:00. This study uses data for mean sea level pressure, zonal and meridional wind components to force a numerical storm surge model, and integrated column vapour and sea surface temperatures for the analysis of differences between AR systems associated with compound events and those without a connection between AR and compound event for the time period from 1979 - 2015.

### 3.2  Atmospheric river database

Information about ARs occurring in the ERA-Interim reanalysis data between 1979 and 2015 is taken from an online AR database, which is based on the algorithm presented in Guan and Waliser (2015) and provided online by Bin Guan. The database contains information about the geometrical shape, axis and landfall locations of ARs in the ERA Interim dataset on a global grid with a spatial resolution of $1.5^o \times 1.5^o$. It further provides the land-sea and coastal mask the detection algorithm used to determine AR landfalls. These masks are equivalent to those used in the IFS release Cy31r2 that generate the ERA-Interim reanalysis data.

ARs in the database are identified using the integrated vapour transport (IVT) spreading pressure levels between 1000 hPa and 300 hPa. If the IVT exceeds both an intensity threshold of its local $85^{th}$ percentile and a minimum of 100 kg m$^{-1}$ s$^{-1}$, the structure has the potential to be classified as an AR. However, only those structures with a length of at least 2000 km and a length to width ration of two or higher are classified as ARs. Atmospheric IVT structures that do not show a significant poleward component are neglected. For more details about the detection of ARs and a validation of the applied detection algorithm the reader is referred to Guan and Waliser (2015).

As mentioned in Section 2 the study presented in this paper isolates ARs in the database that made landfall along the western north coast of the European mainland, i.e. at the French, Belgium, Dutch and German North Sea coasts. This choice is based on the potential of ARs to affect large geographical regions due to their geometric characteristics. All assessments are limited to the impact of these AR systems on the Dutch Delta.

### 3.3  E-OBS precipitation dataset

The E-OBS precipitation dataset provides information of daily precipitation sums over Europe (land only) and spans over a time period from 1950 until present. It is derived from observations at stations across Europe and maps precipitation on a





variety of spherical and regular grids. For a detail description of the data set the reader is referred to Haylock et al. (2008). In this study we use data on a regular grid with a $0.25^o$ resolution. The time period taken into consideration is equivalent to the one used for the generation of the AR database described in the previous section, i.e. 1979 to 2015 (Section 3.2). Precipitation sums for the different regions under investigation have been derived by isolating precipitation data over the grid boxes within

the region SNL, LIJ and NENL as indicated in Fig. 1.

### 3.4 The Storm Surge Model WAQUA/DCSMv5

In this study water levels along the Dutch coast are determined using the Dutch continental shelf model WAQUA/DCSMv5 (hereafter WAQUA; Gerritsen et al., 1995). Based on the two dimensional shallow water equations, WAQUA calculates water levels in the North Sea basin taking into account sea level pressure, 10-meter wind speeds and the astronomical tide at the

domain boundaries using ten harmonic constituents. For selected stations along the coast, WAQUA provides local water level time series with a 10-minute frequency. The output further contains information about the contribution of the tidal component and non-tidal residual (hereafter referred to as surge) to the total water level at each station.

The meteorological fields driving WAQUA in this study are mean sea level pressure and 10-meter wind fields from the ERA-Interim reanalysis database (Section 3.1). In this set up, WAQUA is able to reproduce observations from gauge stations rea-

sonably well (e.g. Sterl et al., 2009; Ridder et al., 2018). However, while generally reliable, WAQUA tends to underestimates extreme water levels, particularly those with long return periods (Ridder et al., 2018). However, this is not considered to be of major significance for this study as the results presented here are based on quantiles and are therefore determined relative to water levels produced within the model. Further, the water levels investigated here lie below the one-year return level, thus the negative bias is not expected to significantly affect the results and conclusions of this study.

### 3.5 Definition of compound events

Since this study uses local precipitation as a proxy for run-off we need to define a temporal constraint for the definition of compound events that allows enough time for the precipitation water to reach the coast and interact with coastal waters. At the same time, we need to exclude more complex hydrodynamic processes that are caused by processes taking place further upstream, i.e. outside of the study area, in large catchment areas. Considering the relatively small catchment areas under

investigation here, a run off time of three days seems reasonable. This three-day period should be sufficient to ensure that run-off and other catchment processes have transported the precipitated water close enough to the coastal area to be able to interact with the coastal water level. If precipitation was to occur several days after a coastal water maximum, the collected water in the catchment would reach the coast after the maximum water levels have already subceeded, i.e. too late to cause the compounding effect under investigation in this study. Similarly, if a coastal water maximum takes place too long after a

high precipitation event, i.e. the time-scale is chosen to be too long, the precipitated water might already be discharged into the sea, again not coinciding with a surge extreme. In this case the local impact would result from one or the other variable in isolation, thus could lead to false positives in the identification of compound events. Also, for long time scales the run off might be contaminated by upstream processes unrelated to the synoptic event causing the local precipitation, e.g. snow melt or



isolated precipitation further upstream unrelated to the synoptic system causing the surge. Furthermore, since this study applies daily precipitation sums the selection of a three-day period also ensures the inclusion of extremes that occur closely around midnight of a selected day that otherwise would be associated to a different day and thus falsely considered to not interact with coastal water levels despite the water.

The choice of a threshold to determine whether or not an event is considered to be "extreme" needs to take into account the limited data availability of daily values in precipitation and only 37 years in water levels. Therefore, we need to select a threshold low enough to deliver a reasonable number of events to allow a solid statistical analysis. At the same time, setting the threshold too low would prevent the assessment of the high tail of the multivariate distribution by including events with only moderate impact that are less relevant for the analysis of compound events. Therefore, we choose a relatively low threshold to

define extreme precipitation and total water levels, namely the 95[th] percentile of the respective variable. The choice of rather weak extremes like this ensures numbers of exceedances sufficient for a solid investigation of the relatively short study period. According to the above argumentation on timing and threshold, in the remainder of this paper, an event is referred to as a compound event (CE) if the centred three-day precipitation sum over one of the chosen regions in the study area exceeds its 95th percentile and the total water level at the associated coastal station exceeds its 95th percentile at any point during the same

three-day period.

## 4 Results

### 4.1 Climatology

Throughout the study period with a total of 13,513 days, roughly 17-19% of days display conditions with an AR located over the Netherlands at atleast one of its four six-hourly reanalysis timesteps (Fig. 2). This shows that ARs are a common

phenomenon in this region with an AR passing over the Netherlands roughly every 3-5 days. The number of compound events in the 37 years of ERA-Interim data ranges from 93 (DLZ) to 106 (HvH) events with the majority of CEs coinciding with the presence of an AR over the Netherlands on the same day or within ±1 days of the event (Table 1). Only a small fraction of 18% (Harlingen) to 28% (Delfzijl) of CEs does not show any association to the presence of an AR over the Netherlands.

Compound events mainly occur in the winter six months (SON and DJF) with a peak in November at HvH and DHR and

in January at HRL and DLZ (Fig. 3). CEs associated with ARs occur almost exclusively during the winter six months with the exception of a few events between one (DLZ) and five (HvH) in March. Due to the small total number of CEs with AR association in March and the lack of CEs in the summer months, the remainder of this study will focus the assessment on the winter six month (SON and DJF) only.

The mean SLP anomaly pattern during all wintertime CEs in the ERA-Interim period shows a distinct difference to the mean

climatological with a pressure dipole rover Europe and the eastern North Atlantic esembling the positive positive phase of the North Atlantic Oscillation (NAO+; Fig. 4a). To allow a thorough investigation of the impact of ARs on CEs in this study, we differentiate three types of CEs, namely those events that co-occur with an AR over the Netherlands, either on the day of the event (hereafter ARCEs; Fig. 4b) or one day before and/or after (hereafter ARpm1dCEs; Fig. 4c), and those that occur





in the absence of an AR in the three days around the event (hereafter noARCEs; Fig. 4d). Since the first two types of CEs show very similar atmospheric climatological anomalies (Fig. 4b and c) we will focus our analysis on the difference between noARCEs and ARCEs only and classify the ARpm1dCEs as only a slight variation of ARCEs. Therefore, all conclusions drawn for ARCEs are qualitatively the same as for ARpm1dCEs. In contrast, the third type of events, i.e. noARCEs, occur
under a significantly different anomaly pattern compared to ARCEs (Fig. 4b and d). While anomalies during ARCEs resemble the overall anomaly pattern of all CEs regardles of AR occurrence (Fig. 4b), the pressure dipole in case of the noARCEs displays a tilted axes stretching from northwest to southeast, thus resembling the pattern of the second mode of variability of the circulation over the North Atlantic, namely the East Atlantic (EA) pattern (Fig. 4d). In the mean climatology, however, the EA pattern is overpowered by the NAO+ dipole due to the large number of CEs with an AR over the Netherlands compared
to those without (Table 1). Thus, only the division of CEs into the two types of ARCEs and noARCEs reveals the EA pattern and allows a comprehensive analysis of the problem. A detailed analysis of the evolution of the SLP anomalies leading to both types of CEs is discussed in Section 4.3.1.

## 4.2  Joint Probability Distribution

To assess the impact of ARs on the correlation between precipitation and coastal water levels at the four study locations,
Fig. 5 compares the joint probability density distributions of three-day precipitation sums and maximum water levels for days without an AR over the Netherlands (hereafter noARdays; black contours) to the same distribution for days with an AR over the Netherlands (hereafter ARdays; red contours). The median of the distribution considering ARdays (red cross) is significantly shifted to higher precipitation sums compared to the median of the noARday distribution (black cross). The shift to higher three-day maximum water levels is slightly less pronounced, but nevertheless clearly visible. This response in the median reflects the
nature of the meteorological phenomenon causing ARs which induces positive precipitation and storm surge anomalies. This can also be seen in the differences between the two distribution with the distribution of ARdays (red contours) reaching further into the part of the graph indicating high precipitation and water levels than the distribution of noARdays (black contours).
In order to understand the influence of ARs on coastal CEs the next step of the analysis focuses on the high tail of the two distributions. For this we select only those days with conditions that fulfil the definition of CE used in this study (grey and
magenta scatter plot in Fig. 5). In this region the medians of the two distributions are almost identical (black and red plus) at Hoek van Holland, Den Helder and Harlingen (Fig. 5a-c). This suggests that the conditions during CEs with and without AR over the Netherland have caused impacts of similar severity in terms of the joint effect of precipitation and water level at these three stations. Only at Delfzijl the two medians differ significantly (Fig. 5d). Here CEs caused by conditions influenced by an AR tend to have a higher impact on precipitation, while storm surge levels seem to be less affected than in the case of
noARCEs.

## 4.3  Difference in meteorological conditions before and during compound events with and without AR association

To determine if ARs significantly alter CEs in the Netherlands this section assesses the conditions during CEs with and without association to ARs. The analysis is focused on CEs at Den Helder. This choice was motivated by the geographical location of





this station close to the Wadden Sea and the fact that the station is situated at the north-western corner of the Dutch coastline. Thus, Den Helder borders the North Sea at two sides, the north and west. As discussed in Section 2, this is not the case for the other stations. Thus, choosing Den Helder as representative ensures that the assessment accounts for synoptic systems moving in from the north as well as from the west. Further, most of the compound events at Den Helder occur in close temporal proximity to compound events at (at least one of) the other stations which makes Den Helder a valuable representative for all four stations when it comes to the occurrence of CEs.

### 4.3.1 Development of sea level pressure and integrated vapour transport

The comparison of the mean anomalies in daily sea level pressure (SLP) and integrated vapour transport (IVT) before and on the day of a CE at Den Helder shows a clear difference in the conditions of CEs with and without association to ARs (Fig. 6 a, e and i; b, f and j, respectively).

CEs associated with ARs (ARCEs) show little temporal variability in their mean anomaly pattern throughout the week before an event (Fig. 6a, e and i). The overall pattern is comparable to climatology with a high-pressure system over the Azores and a low-pressure system in the North, in this case stretching from the east of Greenland and the Norwegian Sea (Fig. 6c, g and k). The evolution of the atmospheric conditions during this time is mainly limited to changes in the amplitude of the sea level pressure features. Thus, the storm track remains unchanged and is comparable to that under the conditions of a positive North Atlantic Oscillation phase. The low-pressure system develops a stronger anomaly than its positive counter part. This hints to the importance of the storm system as a driving mechanism in the ARCEs. The horizontal dipole that the two pressure systems build and is typical for the North Atlantic Oscillation (NAO), guides the IVT through a small corridor over the UK the north of France before hitting the Netherlands further inland. Therefore, ARs making landfall in the UK, France and the Netherlands itself have the potential to be part of the synoptic system that causes a CEs (Fig. A1).

In contrast, compound events without the involvement of ARs (noARCEs) the spatial SLP anomaly patterns vary strongly with time during the week before the event (Fig. 6b, f and j). Seven days before the event, SLP anomalies show two moderate positive maxima, one stretching from Greenland to east of Iceland and one off the coast of the UK (Fig. 6b). The first anomaly maximum is caused by a high-pressure system over Greenland; the latter by a high-pressure system over Spain stretching further north than the Azores high under normal conditions (Fig. 6d). Over the following few days the high-pressure system over Greenland and Iceland temporarily weakens and a low-pressure system moves in from the west (Fig. 6h) leading to a moderate negative SLP anomaly north of $60°N$ (Fig. 6f). At the same time the high-pressure system over Spain merges with a high-pressure system moving in from the western Atlantic (Figures 6d and h). This causes a strengthening of the positive anomaly west of the UK and increases its extent to cover large parts of Western Europe, Scandinavia and an area over the North Atlantic between $40°N$ and $60°N$ reaching up to $55°W$ (Fig. 6f). On the last days before the CE the Azores high moves back westward ending up in a position that is slightly further north than under normal conditions (Fig. 6l). At the same time, the low-pressure system in the north of the Azores high moves eastwards towards Scandinavia, and the high-pressure system over Greenland strengthens to contribute to the strong positive anomaly seen over most of the northern North Atlantic on the day of the CE. The resulting anomaly pattern resembles the anomalous conditions of the East Atlantic (EA) pattern, the second



mode of interannual varibility over the North Atlantic (Barnston and Livezey, 1987; Comas-Bru and McDermott, 2014). In this position the Azores high acts as an atmospheric blocking system together with the high-pressure system over Greenland that detains the negative pressure anomaly over Scandinavia (Fig. 6j). These conditions cause a northwards excursion of the storm track. In turn, the resulting meandering of the storm track causes winds and the IVT to hit the Dutch coast with

a stronger northerly component than normal conditions. Thus, the conditions during noARCEs favour high surge levels and higher precipitation along the northward facing European coastlines. The reason that the increased IVT in the case of noARCEs cannot be classified as ARs, even if they occurred in long filaments fulfilling the geometric definition of the AR definition, lies in exactly this change of the storm track which forces the IVT to enter the North Sea basin from the north. As a result, the IVT, while significantly increased compared to climatology and in its absolut values comparable to the IVT during ARCEs,

lacks a distinct poleward component, which is one of the crucial characteristics of ARs according to their most commonly used definition.

### 4.3.2 Development of Sea Surface Temperature (SST) anomalies

The changes in SLP conditions are also reflected in the anomalies in sea surface temperature (Fig. 7) through the connection between surface winds and ocean currents. In the case of ARCEs, SSTs respond to conditions that induce gradual, spatially

consistent changes due to the small spatial variability of the SLP anomalies in this case as discussed in the previous section (Sec. 4.3.1). As a result, the wind anomalies, which increase with time getting stronger closer to the event (Fig. A2), induce an increase in SSTs that covers large parts of the western (tropical and subtropical) North Atlantic, the North Sea and parts of the Norwegian Sea on the day of the event (Fig. 7a, c and e). The most important difference to noARCEs is the mean positive SST anomaly off the east coast of North America that, in case of ARCEs, is persistent throughout the week before an event.

This positive anomaly is most likely maintained through the increasing transport of warm tropical waters into the midlatitudes through a strenghtening of the north north-easerly component of the wind field throughout the week before an event. However, a detailed account on the driving mechanisms behind the response of SST can only be obtained by an in depth analysis of the complex interplay of changes in Ekman transport, upwelling/downwelling and ocean-atmosphere heat exchange which is beyond the scope of this paper.

In contrast, SLP anomalies during noARCEs, and with this the anomalies in 10-m wind fields (Fig. A2), over the North Atlantic trigger a warm anomaly in the midlatitudes, including the North Sea, and subtropics, with a negative anomaly in the tropics north of the equator and north of 60°N (Fig. 7b, d and f). Interesting is the negative SST anomaly in the region where ARs that hit Europe generally originate from, i.e. the tropical North Atlantic. This negative SST anomaly stretches from coast to coast around the 20°N latitude with a strong maximum in the upwelling region off West Africa. While this large-scale SST

anomaly pattern broadly remains persistent throughout the week before an event, the changing conditions leading to noARCEs alter local SSTs through a variety of mechanisms resulting in the positive SST anomaly to dissappear in the substropical and western part of the North Atlantic (Fig. 7f). This is mainly driven by alterations in the Ekman transport across the basin. During the shift of the SLP anomaly pattern from conditions resembling the negative phase of the NAO towards an EA-like pattern closer to the event, the tilt of the North Atlantic pressure dipole over changes. This induces alterations in wind conditons, which



in turn lead in a flow of cold water from the northeast to the southwest. As a result a cold anomaly off the east coast of North America develops and the warm SST anomly in the subtropical North Atlantic contracts to the eastern Atlantic.

### 4.3.3 Development of precipitation anomaly patterns

The mean anomalies in precipitation during noARCEs and ARCEs reflect the differences in IVT between the two cases. For ARCEs, precipitation anomalies occurr on a much larger scale than in the case of noARCEs (Fig. 8). Additionally they show a strong positive anomaly in central Europe reaching as far south as the Alps and far into the east (Fig. 8b). Further noteworthy are the strong mean positive precipitation anomalies in Northern Ireland and along the west coast of the UK. Together with the increase in mean precipitation over northern France, this pattern reflects the mean direction with which the IVT and thus the ARs, are moving over Europe. As mentioned earlier when discussing SLP and IVT conditions during ARCEs, the IVT is travelling more zonally before and during ARCEs. Therefore, ARs have the opportunity to affect larger regions and thus explaining the large-scale precipitation anomalies under these conditions. For instance, the path over land of an air mass travelling zonally over the UK is much shorter than that of its counterpart travelling in a meridional direction and thus crossing the full latitudinal extent of the landmass. As a result the air mass travelling from west to east tends to loose less moisture through precipitation. Further, the air mass has the opportunity to replenish lost moisture on its way over the North Sea or while travelling along the English Channel before making landfall on the European mainland and precipitating the rest of its moisture there.

As mentioned earlier, the IVT in the case of noARCEs tends to contain a stronger than normal northerly component which causes them to hit the Netherlands almost straight from the North due to the EA-like pattern of the prevailing mean SLP anomalies (Fig. 6i). Accordingly, the precipitation anomalies reflect this by exhibiting a positive anomaly along the Dutch coast (Fig. 8a). On their way over land the moisture lost through precipitation cannot be replenished as easily as over water which leads to quick drop-off in precipitation southwards of the coastline with the majority of precipitation being dropped north of 50°N. A similar anomaly pattern can also be seen in the north of the UK, where the same mechanism influences precipitation. This results in very localised precipitation anomalies in the northern most regions of Northern Europe. The north coast of France, however, which is located in the lee of the UK in the case of noARCEs, shows hardly any anomalous precipitation as water vapour is removed through precipitation over the north of the UK and the English Channel not being sufficiently wide for the moisture to be replenished.

### 4.4 Difference AR with CE and those without

In order to be able to exploit the potential of AR systems to predict coastal CEs, this section assess the differences in atmospheric and oceanic conditions of AR systems with association to CEs (hereafter CEARs) and those without CEs (hereafter noCEARs). For the comparison of anomalies between the two types of ARs we focus on the days with an AR over the Netherlands. Here, the mean anomalies in daily SLP, IVT, SST and precipitation in the case of noCEARs (Fig. 9) are significantly less pronounced than those during CEARs (Fig. 6i, 7e and 8b). This is based on the fact that the mean changes in SLP for noCEARs are not strong enough to create a significant dipole pattern (Fig. 9a). While the mean negative anomaly over the



north of the UK is well established, there is no mean positive anomaly in the location of the Azores High comparable to that evolving in case of CEARs (Fig. 6i). This indicates that the position and strength of the Azores high plays a major role in the determination of whether an AR system can lead to a coastal CE or not.

As a result of the lack of a mean dipole structure, mean wind fields during noARCEs do not produce a consistent change in
surface ocean circulation and thus do not show a strong mean SST anomaly pattern. The same is true for precipitation. This suggests, that overall only strong AR systems, consisting of a strong SLP dipole and carrying high moisture amounts, have co-incided with the occurrence of compound events in the Netherlands. However, this does not mean that all strong AR systems, i.e. those with strong SLP anomalies, have been associated with compound events since the mean in the noCEARs case is derived from a much higher number of events compared to the CEARs case. Thus some strong AR systems might have failed
to induce sufficient precipitation due to the lack of air moisture or the necessary wind conditions in terms of wind direction to induce a compound event at the Dutch coast.

## 5 Discussion

This study presents a first classification of coastal CEs by using one specific atmospheric phenomenon as a base resulting in two types of CEs, i.e. (i) events with AR involvement and (ii) events without. This classification can be used to determine
the focus of future assessments and deepen the analysis of the driving processes of coastal CEs consisting of heavy precipita-tion and high coastal water levels. While other coastal CEs might require different categories based on other climatic or even socio-economic factors, the here presented choice of ARs as determining factor is the most suitable considering the purpose of this study, i.e. the investigation of the impact of ARs on coastal CEs in the Netherlands. Thus, it was possible to identify conditions leading to CEs that do not involve AR. These would have been masked if the analysis had only taken into account
the mean conditions during CEs which are dominated by the large relative number of CEs with AR involvement. While these atmospheric conditions have been known to potentially cause hazardous conditions for the Netherlands and thus have already been thoroughly studied, conditions with the Azores High acting as blocking system, as realised during the second type of coastal CE, have gotten little attention. With the findings of this study we provide an impulse to extend future investigations into this direction.

Further, by identifying large-scale atmospheric conditions that lead to coastal CEs and comparing them to similar conditions with low impact we provide a tool for the early identification of possible compound events. This is particularly useful in the light of the higher predictability of large-scale atmospheric features, such as SLP patterns and atmospheric moisture content, compared to small-scale events, such as precipitation and wind extremes (Lavers et al., 2014). Therefore, the results of this study could be used for the early identification of compound events that have the potential to cause disruptive impacts in the
Netherlands and thus allow an early warning of up to one week in advance.

While this study focused on local precipitation rather than river discharge, we show that the presence of ARs leads to precipi-tation anomalies that cover large areas of the Rhine catchment. This indicates that, additionally to the chance of the occurrence of CEs in the form of heavy precipitaiton and high surge, it is likely that ARs are also linked to the co-occurrence of high





surge levels and extreme river discharge. These two hazards have been shown to be correlated at a time-lag of several days with storm surge extremes preceding high river discharge (Klerk et al., 2015; Khanal et al., 2018). Our results are in agreement with this, taking into account the time it takes for hydrological processes to transform precipitation over a large catchment into river discharge at the coast or further downstream. As a result, it is possible that ARs aggravate coastal flood risk even further

by causing extreme river discharge closely after a compound event consisting of heavy precipitation and high coastal water levels. We leave the investigation of the existence of a statistical connection between these two occurrences and the possible implications for local flood risk to future studies as this falls outside the scope of the work presented here.

We acknowledge that our findings are based on model results and observations and the used data contains the known biases and shortcomings associated with the respective data source. However, the impact of data biases is unlikely to affect the qualitative

statements made in this study as most results are based on quantiles, thus dampening the effect of possible biases present in the used datasets.

We also note that the AR-detection algorithm is determined by a specific definition of ARs. The particular algorithm applied here was chosen due to the fact that this study was motivated by the work of Waliser and Guan (2017). While the application of other algorithms might introduce some variations in the results of this study, their effect would be marginal and is not expected

to significantly change the conclusions of this study.

However, we would like to remark that our analysis of the conditions during the second type of CEs, namely noARCEs, highlights a limitation of the generally accepted condition often used in AR detection algorithms which excludes IVT structures that lack a significant poleward component. We have shown that during noARCEs the IVT reaching the Dutch mainland is significantly incrased with absolute values comparable to those in the case of ARCEs. Further, we have demonstrated that both

types of CEs lead to comparable impacts in terms of precipitation regardless of the inclusion of the underlying IVT structure into the AR catalogue or not. While it is possible that some of the IVT structures during noARCEs were discarded due to the applied geometric constraints, some ARs only failing the poleward transport condition might have been falsely excluded. It is therefore possible that ARs play a much more important role in the occurrence of CEs than identified in this study. We thus suggest that excluding IVT patterns from an AR catalogue based on their poleward transport, could lead to an underestimation

of the risk that ARs pose for coastal regions. Further, if poleward transport should no longer be considered as a detection criterion for ARs, the classification made in this paper of CEs into three types (ARCEs, ARpm1dCEs and noARCEs) might need to be extended accordingly. Therefore, we advise to apply the AR classification criterion requiering an IVT object to have a considerable poleward component with care and its implications kept in mind when assessing the influence of ARs on CEs.

Nevertheless, our study provides a valuable extension of our understanding of costal CEs and their driving mechanism at one

specific geographic location focusing on one particular atmospheric phenomenon. With this, we hope to inspire future work to extend our assessment to include the impact of other phenomena to complement the results of this study. Further, we encourage to apply this and similar assessments to other geographical regions to elaborate on differences in the importance of drivers under different climatological conditions and identify other equally important atmospheric phenomena influencing coastal and other CEs.



## 6 Summary and conclusions

In this study we used the output of a numerical storm surge model (WAQUA/DCSMv5) and observed precipitation data (E-OBS) throughout the ERA-Interim period (1979-2015) to assess the role of atmospheric rivers in the occurrence of compound events consisting of heavy precipitation and high coastal water levels at four stations along the Dutch coast. Our results show

that the majority of past compound events have been associated with the presence of an AR over the Netherlands. Further, we demonstrate that days with an AR over the Netherlands tend to be wetter and have higher water levels than those without. However, this is not realised in the high tail of the joint distribution of the two variables, where the impact of ARs fails to significantly affect the median of the joint distribution (with the exception of Delfzijl). From this we conclude that, while ARs play an important role in the occurrence of compound events, their mean impact is comparable to that of events without AR

involvement. Nevertheless, the introduced classification of compound events into two categories, (i) events with AR influence caused by a NAO-like SLP anomaly pattern and (ii) events without AR influence occurring under EA-like SLP anomaly conditions, shows to be useful in order to isolate atmospheric patterns of events that are otherwise masked by the dominance of the number of compound events with AR involvement. Further, in combination with the mean SST anomaly patterns and the NAO- and EA-like SLP patterns specific to each type of event that we identified here, we provide vital information for the

possibility to predict compound events. As shown in this study, climatological anomalies leading to the two types of coastal CE are visible at least seven days in advance of an event. It is thus possible to include the atmospheric and oceanographic features leading to CEs that have been identified in this study as indicators in an early warning system for possibly hazardous conditions along the Dutch coast.

*Data availability.* The data for AR characteristics used in this study are made available by Bin Guan and obtained from https://ucla.box.
com/ARcatalog.

*Author contributions.* NR had the initial idea for the study, performed the surge model experiment, analysed the data and authored the manuscript. HV and SD provided suggestions for analytical metrics and commented on the manuscript.

*Competing interests.* There are no competing interests present.

*Acknowledgements.* The authors would like to thank B. Guan and D. Waliser for providing access to the used AR database and shar-
ing their AR-detection algorithm. We also acknowledge the E-OBS dataset from the EU-FP6 project ENSEMBLES (http://ensembles-eu.metoffice.com) and the data providers in the ECA&D project (http://www.ecad.eu). This study was funded by the Netherlands Organisation for Scientific Research (NWO) as part of the project "Impacted by Coincident Weather Extremes" (ICOWEX; grand number 869.15.017).



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



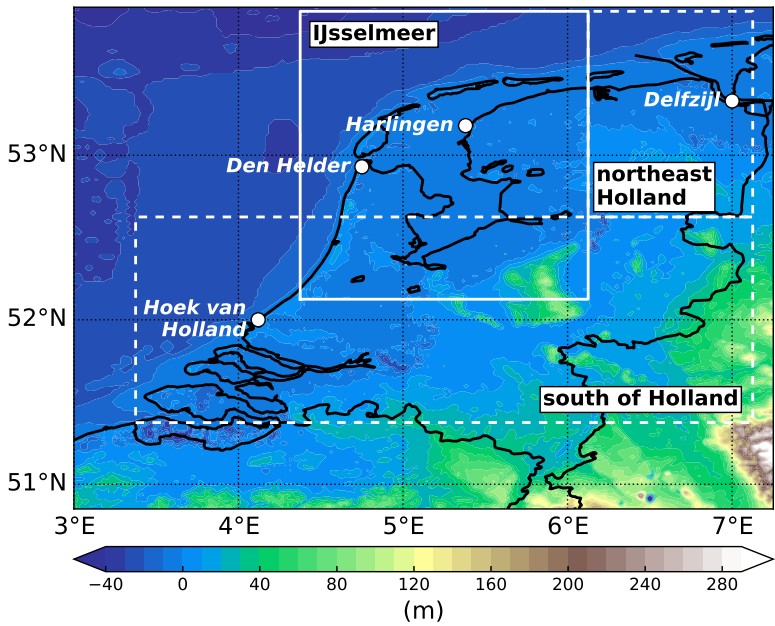

**Figure 1.** Topographical map of the study area showing the geographic location of the four coastal stations under investigation in this study. White boxes indicate the three regions that were considered for local precipitation, i.e. south of Holland for Hoek van Holland, IJsselmeer for Den Helder and Harlingen and northeast Holland for Delfzijl. Elevation data is derived from the ETOPO1 Global Relief Model(Amante and Eakins, 2009).

**Table 1.** Number of compound events associated with AR landfall relative to the total number of compound events at the four coastal stations under investigation during the ERA-Interim period.

|  | Hoek van Holland | Den Helder | Harlingen | Delfzijl |
| --- | --- | --- | --- | --- |
| all CEs (full year) | 106 | 93 | 99 | 93 |
| winter CEs | 93 | 86 | 90 | 89 |
| CEs with AR over NL (winter only): |  |  |  |  |
|   - on day of event | 43 | 49 | 52 | 38 |
|   - 1 day before or after event | 28 | 21 | 23 | 28 |
| CEs without AR over NL (winter) | 22 | 16 | 15 | 23 |



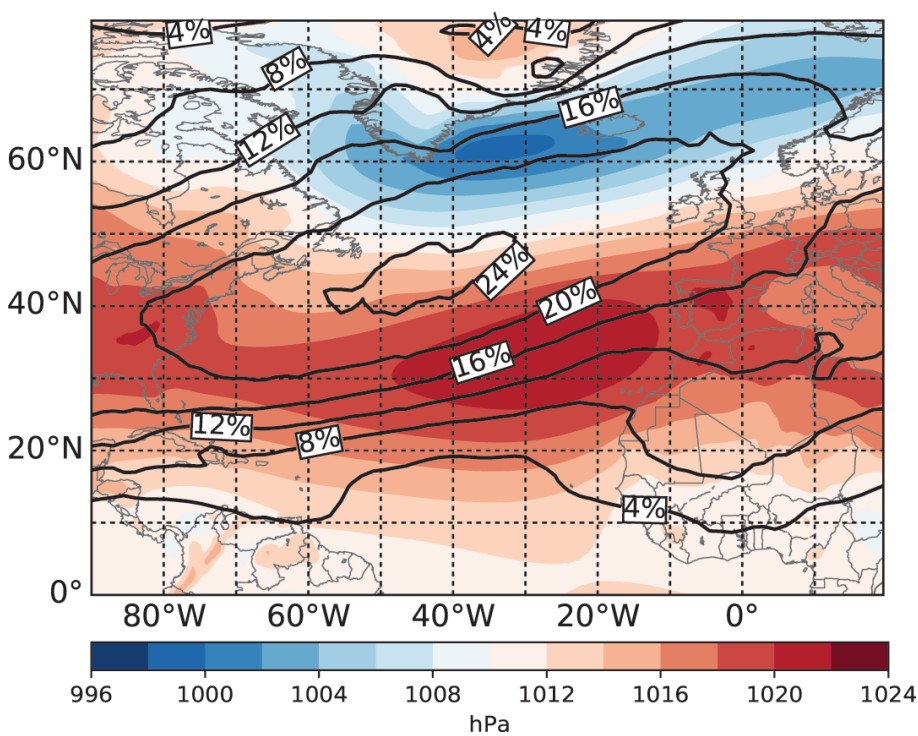

**Figure 2.** Climatology of daily mean sea level pressure (SLP; colour shading) and areas covered by AR (contour) throughout the study period (1979-2015).





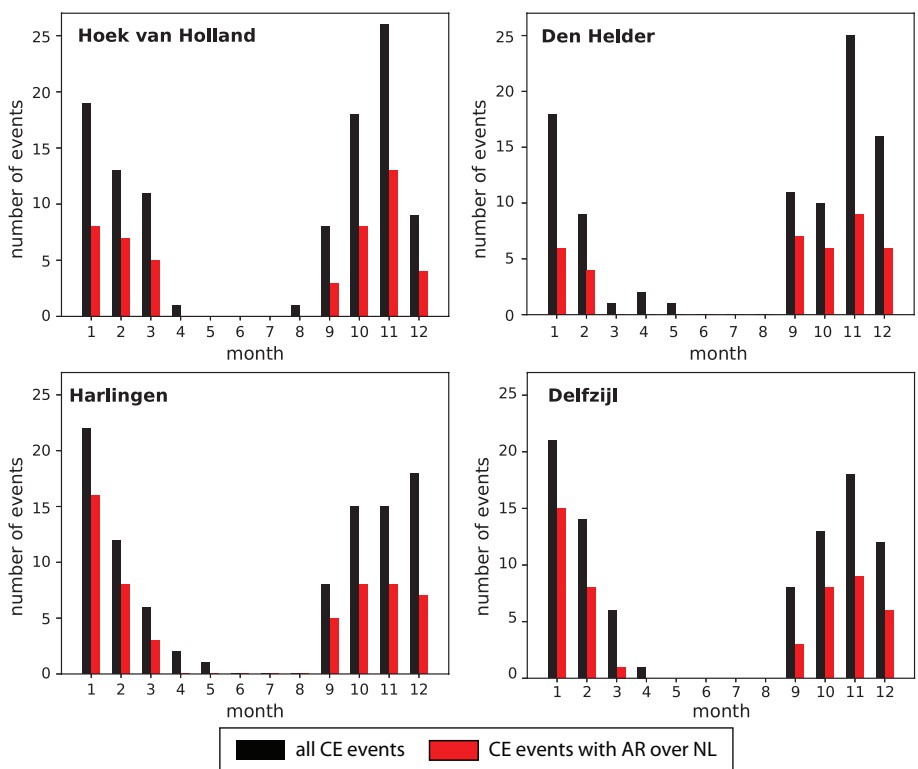

**Figure 3.** Number of compound events per month at the four coastal stations assessed in this study. Black columns indicate the number of all CEs (ARCEs + noARCEs + ARpm1dayCEs), while red bars show the number of CEs with association to an AR over the Netherlands (ARCEs).



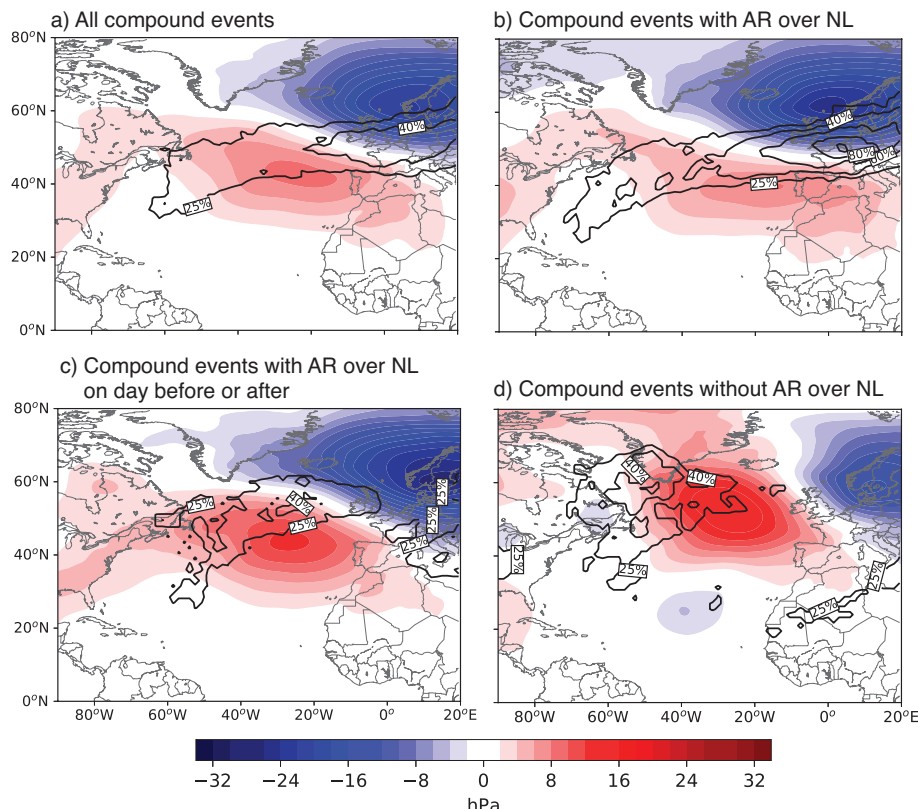

**Figure 4.** a) Mean anomalies in daily mean sea level pressure (colour shading) and relative area covered by ARs during all compound events at Den Helder during the study period. b), c) and d) as a) but for CEs occurring on days (b) with an AR over the Netherlands, (c) one day before or after a day with an AR over the Netherlands and (d) without AR over Netherlands within a three day period centred around the event.





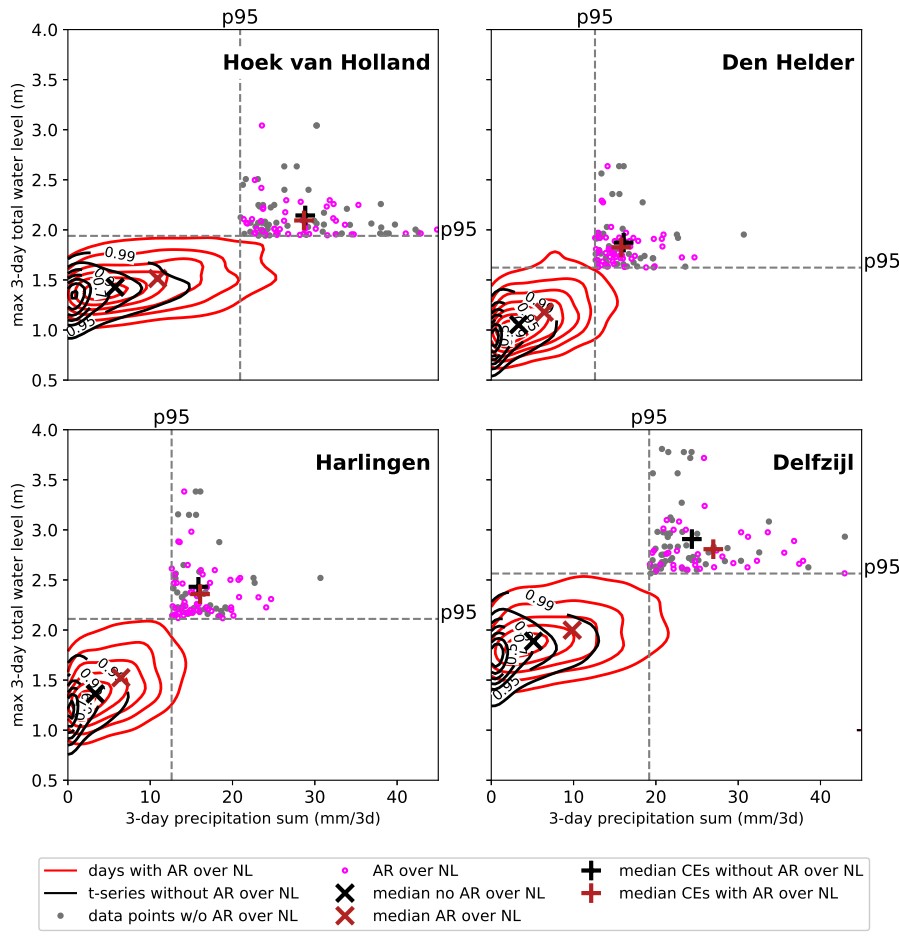

**Figure 5.** Joint probability distribution of three-day precipitation sums (mm) and three-day maximum total water level (m) for days without an AR over the Netherlands (dark contours) and days with an AR over the Netherlands (red, solid contours). Scatter plot in the in the upper right corner of each subfigure show total water level and precipitation pairs with values higher than the 95th percentile of both variables, i.e. compound events. Compound events without an AR over the Netherlands are shown in black, those with an AR over the Netherlands are red. Crosses indicate the position of the mean of the full time series, while plusses show the median of the two variables only taking into account data from CEs. The colour coding for both markers is the same as for the contours.




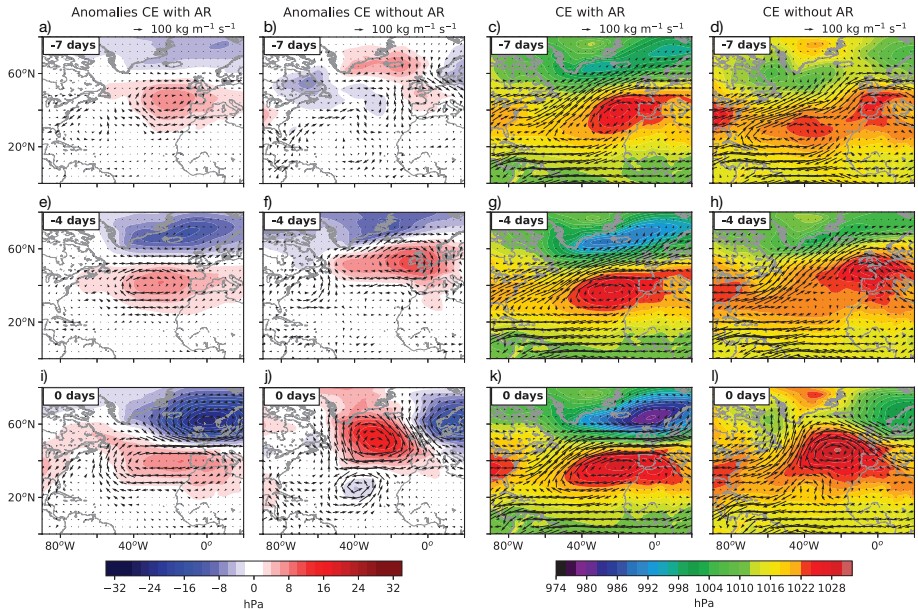

**Figure 6.** Temporal evolution of mean conditions seven (a-d) and four days (e-h) before a CE and on the day of the event itself (i-l). The right two columns, i.e. panels a, e, i and b, f, j, show the evolution of anomalies in SLP (colour shading) and IVT (vector field) during CEs with and without AR association, respectively. The right two columns, i.e. panels c, g, k and d, h, l, show the same but for absolute values of daily mean SLP and IVT.




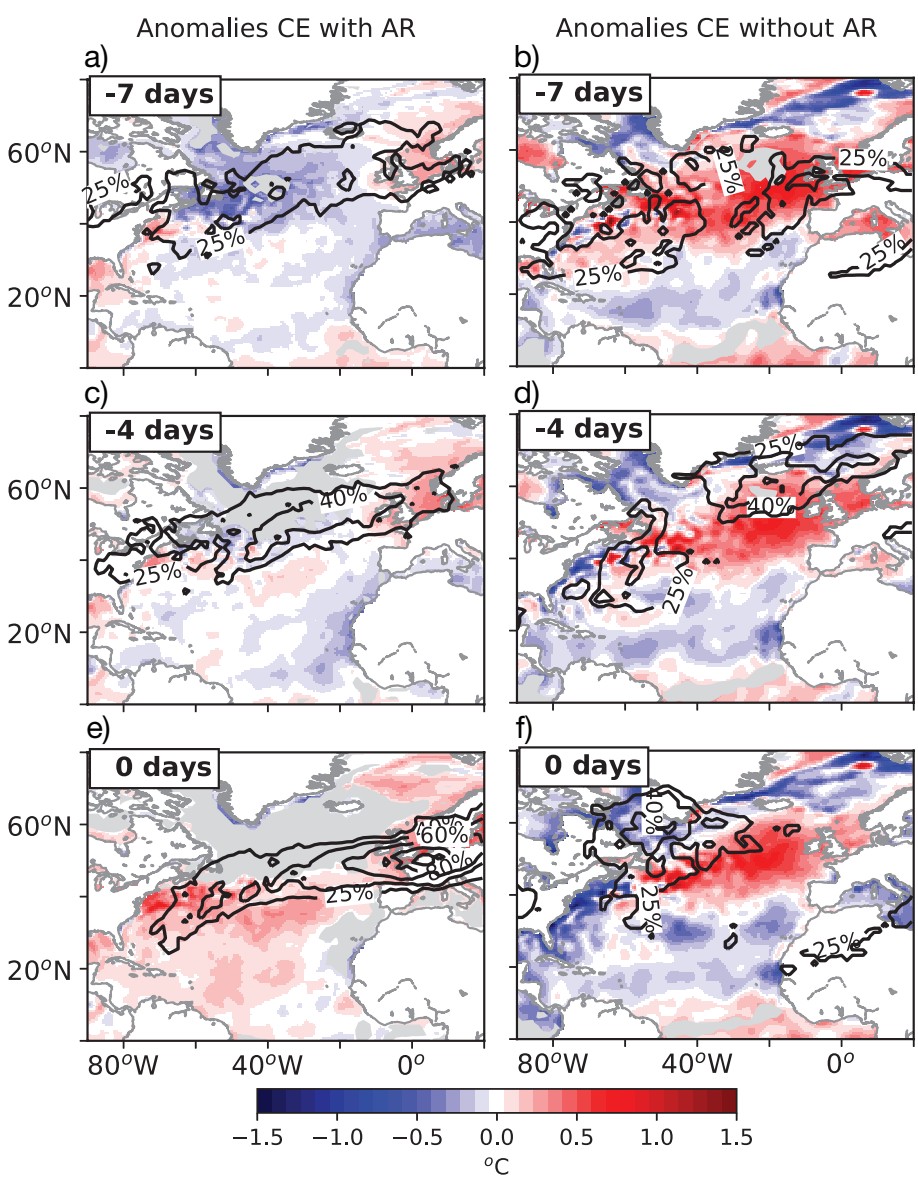

**Figure 7.** AR location (contours) and anomalies in daily mean SSTs (shading) related to CEs with (left panels a, c and e) and CEs without AR association (right panels b, d and f) seven and four days before a CE (a and b; c and d, respectively) and on the day of the event (e and f). Grey areas mark regions with a p-value below 0.05 derived from student t-test of daily mean SST values.





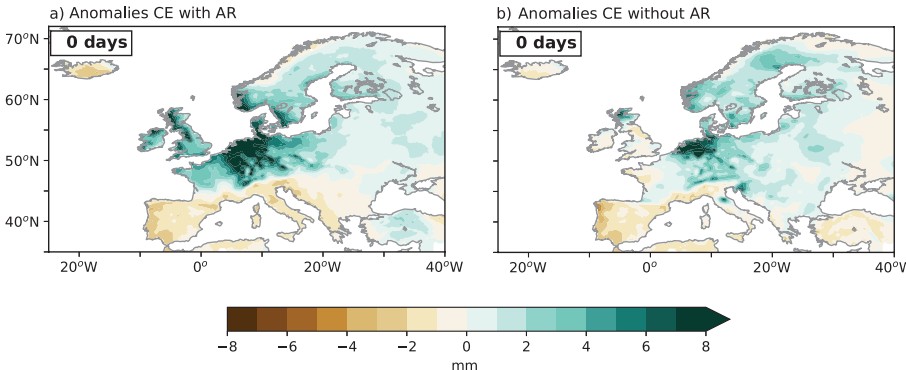

**Figure 8.** Anomalies of daily mean precipitation sums during CEs with (a) and without (b) AR association.

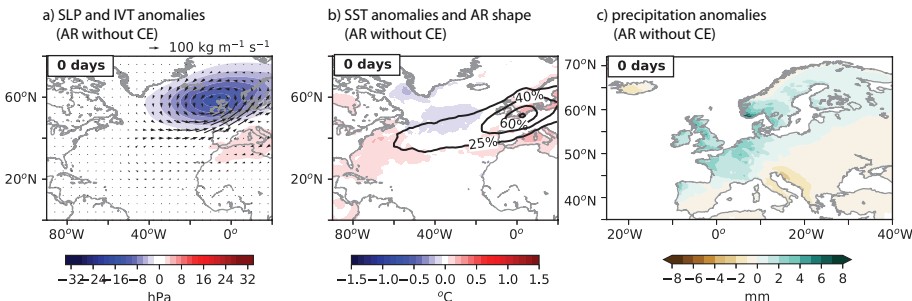

**Figure 9.** Anomalies of (a) SLP (colour shading) and IVT (vectors), (b) SST (colour shading) and relative number of ARs covering an area, and (c) precipitation on days with an AR over the Netherlands without the occurrence of a CE.





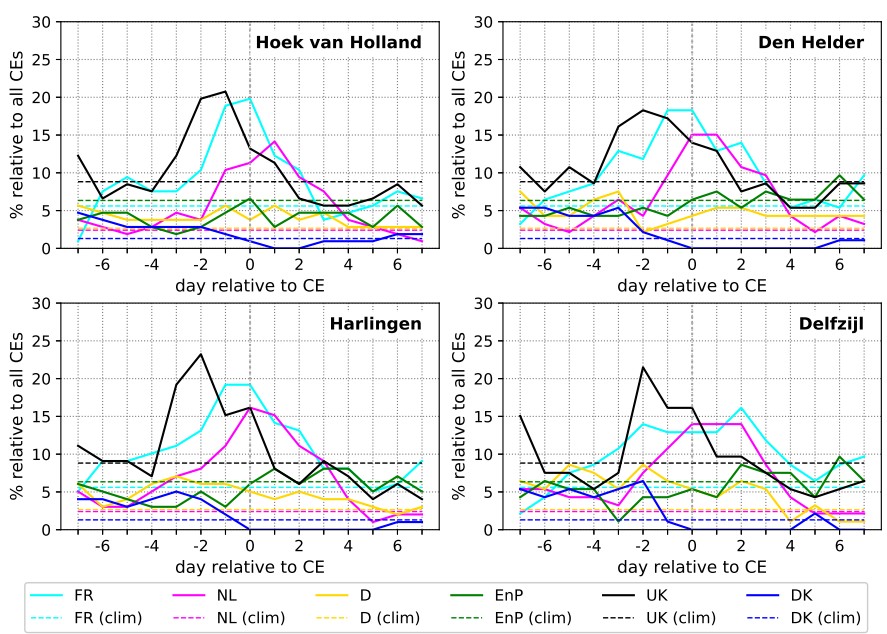

**Figure A1.** Relative number of compound events associated with AR landfall in the UK (black), France (cyan), the Netherlands (magenta), Germany (yellow), Denmark (blue) and Spain and Portugal (green) at a selection of days before and after an event at HvH (upper left), DHR (upper right), HRL (lower left) and DLZ (lower right). Dashed horizontal lines indicate the numbers for days with everyday conditions in the full time series and associate it to the respective landfall location.


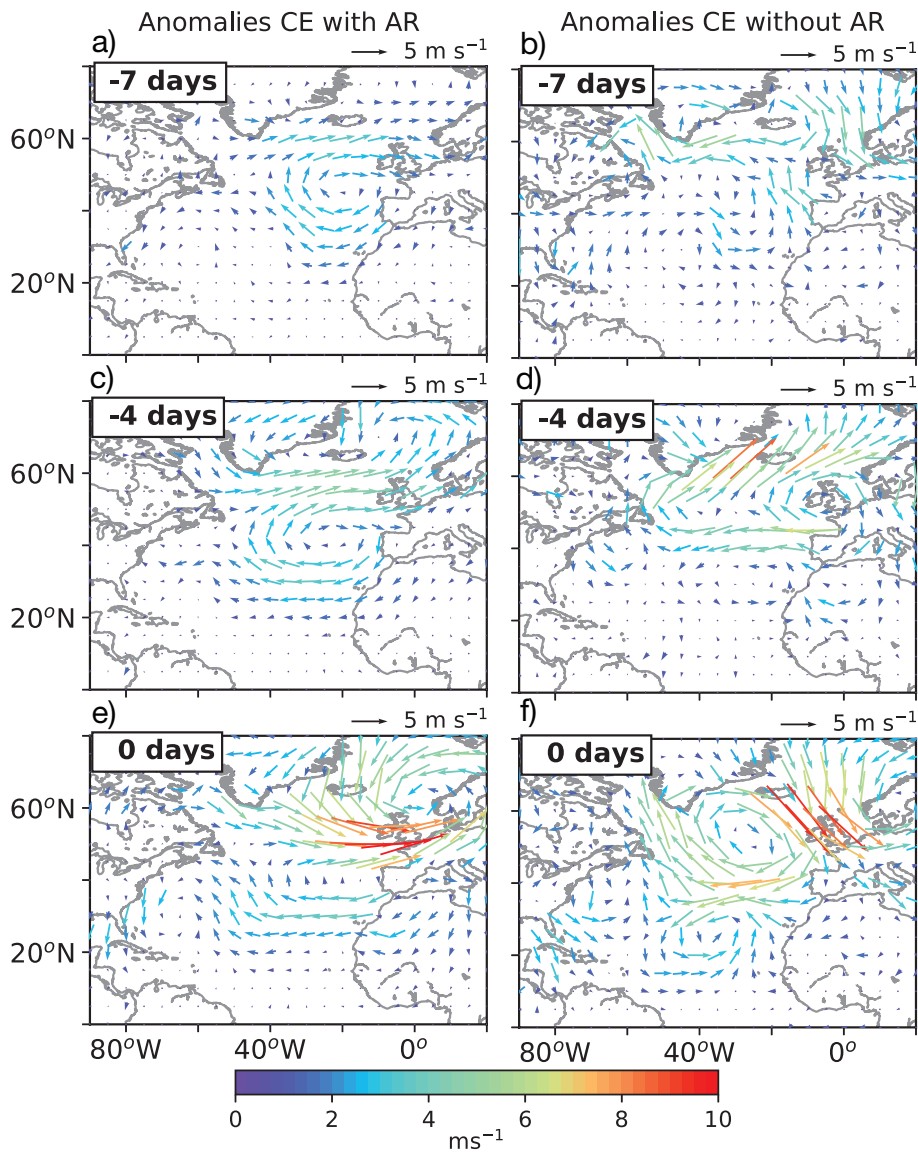

**Figure A2.** Anomalies in daily mean 10-m wind fields related to CEs with (left panels a, c and e) and CEs without AR association (right panels b, d and f) seven and four days before a CE (a and b; c and d, respectively) and on the day of the event (e and f).