# Peer review of "The Role of Atmospheric Rivers in compound events consisting of heavy precipitation and high storm surges along the Dutch coast"

_Natural Hazards and Earth System Sciences, 2018_

## Referee Comment (RC1) · Anonymous Referee #1 · 9 Aug 2018

This paper presents a novel analysis of the association between atmospheric rivers (ARs) and compound events (concurrent high precipitation and high sea water level) along the Dutch coast. The study represents a step further to understand the impacts of ARs beyond the traditional focus on precipitation alone, and may help extend the consideration of ARs in situational awareness and forecast of extreme events to regions where ARs have received relatively less attention in the science and/or applications community.

The analysis procedures are sound for the most part, but needs improvement/amendment as described in my specific comments below. A major missing component is a robust accounting of the statistical significance in the differences between CEs with and without ARs, and between ARs with and without CEs. In the only case where significance test is conducted (Figure 7), the test results do not seem to make physical sense (see specific comments below), which makes me worry about whether the significance test was properly conducted.

Specific comments:

Near Line 5: "accompanied by the presence of an AR", "up to seven days before": does this mean an event is considered AR-accompanied if an AR is present up to seven days before the event? In any case, it would be useful to define "accompanied by an AR".

Near line 10: "local ARs": it is not totally clear what "local" means here. Some ARs travel a longer distance than other ARs, but I'm sure that's what "local" aims to convey here.

Near line 5: "sever": typo of "severe".

Near line 5: "future development of future flood risk": awkward construction.

Near line 15: "in relation with extra-tropical cyclones": it would be more consistent with the definition in AMS Glossary of Meteorology to say "typically in relation with . . ."

Near line 15: "400 - 600 km": add a reference for the quantitative description, or make it qualitative with something like "several hundred km".

Near line 25: "a characteristic not previously assessed": change to something like ", a characteristic not previously assessed for ARs affecting the Europe", because there's at least one study that has examined the effect of ARs on sea water level in western US; see https://agupubs.onlinelibrary.wiley.com/doi/abstract/10.1002/2016GL070086

Near line 30: "projected frequency enhancement and intensification of ARs": Espinoza et al. 2018 could also be cited here to support this statement where they systematically examined and compared such changes across the globe; see

https://agupubs.onlinelibrary.wiley.com/doi/10.1029/2017GL076968

Near line 5: change "on both, " to "on both".

Near line 10: change "both, precipitation and water level," to "both precipitation and water level".

Near line 10: "identify days with the presence of an AR": for the sake of symmetry with CEs, a brief, high-level description of how AR days are identified is warranted here, i.e., based on certain quantile thresholds on intensity and geometry?

Near line 20: change "namely" to "namely,".

Near line 15: "and provided online by Bin Guan": consider removing as the information like this should be (and already is) in the acknowledgement section.

Near line 25: Guan et al. (2018) could also be cited here which provides more validation of the AR database based on comparing to field observations; see https://journals.ametsoc.org/doi/abs/10.1175/JHM-D-17-0114.1

Near line 15: "centred three-day precipitation": I have difficulty understanding what "centred" conveys in this sentence. That is, the word seems unnecessary. If precipitation amounts on day 1, 2, and 3 are a, b, and c mm, respectively, the 3-day precipitation is simply a+b+c mm, i.e., there's no "centering" needed to be done in the calculation.

Near line 20: "number of compound events": the numbers are not fully meaningful without first defining what an "event" is, i.e., is an event counted as a day, a 3-day period, or a continuous period >=3 days?

Near line 20: "within +-1 days of the event": Now I sort of understand what "centred" meant in the earlier sentence. In the example I gave above, does it mean the resulting value of a+b+c is assigned to day 2, and the 3-day period centered on day-2 is considered AR-related if an AR occurred on one or more days of day 1, 2, or 3? Please use the answer to make clarifications in the data section in terms of how a CE is defined,

how an "event" is counted (e.g., if a CE lasted 6 continuous days, is it counted as one event, 2 events, or 6 events?), when a CE is considered to be AR-related or not AR-related, what "day of event" means, etc. Without clear and unambiguous definitions of terms, the statistics presented are hard to make sense of.

Near line 30: "climatological" is a typo of "climatology", and "esembling" a typo of "resembling".

Near line 15: "probability density": for a probability density function, if the function is integrated over all possibilities, the result should be one. But that does not appear to be the case in Figure 5. If you integrate the values over the x-y plane in Figure 5, what does the resulting number represent? That determines how the values contoured in the figure should be called.

Near line 20: add "for" in front of "compound events without".

Near line 10: "absolut" is a typo of "absolute".

Near line 15: "persistent throughout the week before an event": this makes me think that there are conditions during the week prior to the AR that favors the development of warm SSTs and the AR, and in that regard the ARCE (AR+CE) perhaps should be emphasized as indicative of the interplay between these conditions, instead of one causing the other.

Near line 15: "loose" is a typo of "lose".

Near line 25: "Difference AR with CE and those without": please fix the grammar.

Near line 5: "noARCEs": did you mean "noCEARs"? This makes think whether there's a better way to name these events that works better for both the authors and readers, because names like noARCEs and noCEARs are just a bit too cryptic, and when used together with names like ARCEs and CEARs (which I think are identical?) they may cause unnecessary confusions to both the authors and the readers. How about something more descriptive like the following: - CEs with ARs - CEs without ARs - ARs with

CEs (identical to CEs with ARs) - ARs without CEs

Near line 30: "early identification of compound events ... one week in advance": to make this statement and, more importantly, to make the main analysis of the paper more compelling, it is recommended to show that precursor conditions during the week leading to the CEs are statistically different than conditions leading to no CEs. It would be convenient to build on Figure 9 for this purpose, i.e., by expanding it to include the week before (similar to Figures 6 and 7), and adding significance test for the difference between "ARs with CEs" and "ARs without CEs". Significance test is also suggested to be added to Figures 6 and 8 and fixed in Figure 7 for the difference between "CEs with ARs" and "CEs without ARs". The paper heavily relies on statistical analysis (as opposed to dynamics-oriented analysis), so a robust accounting of the statistics is highly desirable.

Near line 10: "a specific definition of ARs": this sounds like there're many different definitions, which I don't think is true. My opinion is that the diversification in AR detection methods (perhaps 20 methods or more exist now) is a manifestation of the difficulty in detecting ARs, not because there're that many different definitions.

Near line 15: "their effect would be marginal": consider removing this statement given the large variations across different AR detection methods (see https://www.geosci-model-dev.net/11/2455/2018/).

Near line 25: change "based on their poleward transport" to "based on their lacking of poleward transport".

Table 1 and where applicable in the text: "on day of event", "one day before or after event": given that the precipitation is a 3-day total, and CEs are defined using a 3-day window, descriptions like these are quite ambiguous. For example, if a CE occurred during the period of January 1-3, then common sense is that "one day before event" is December 31, and "one day after event" is "January 4". But that doesn't seem to be what the authors intended in indicate here. Again, an unambiguous definition of terms

is needed to avoid potential confusions of this kind, as also suggested earlier.

Figure 2 caption: please define "area covered by AR", or how it was calculated. Area would have units of m^2, but it doesn't seem to be the case here. Did you mean AR frequency of occurrence (percent of time steps)? The latter is a more widely used and understood terminology in at least the AR community.

Figure 3: "over NL": what does NL refer to or is it defined somewhere? Are the numbers per single month (i.e., the climatological mean), or the total over the given month? Suppose something happens 3 times in January, and is repeated for the past 100 years, it is more sensible to say it happens 3 times per month, instead of 300 times per month, right?

Figure 6: the plots and fonts are too small. Also, the caption says "The right two columns" twice, the first one of which should be "The left two columns".

Figures 6, 7, 8: "Anomalies CE with AR" etc.: please change to "Anomalies during CEs with ARs", etc. for clarity.

Figure 7 caption: "Grey areas mark regions with a p-value below 0.05": given that a small p-value indicates high significance, do you mean the grey areas are where the values are significant, and the color shadings are where the values are NOT significant? That makes no sense because that would mean you are highlighting the non-significant values, and obscuring the significant values. Also, it is against intuition that the strongest anomaly values (darkest shading in the figure) are with large p-values, i.e., non-significant.

---

## Referee Comment (RC2) · Anonymous Referee #2 · 27 Aug 2018

**Interactive comments on "The Role of Atmospheric Rivers in compound events consisting of heavy precipitation and high storm surges along the Dutuch coast" by Nina Ridder et al.**

**Referee #2**

This study provides an analysis of the association between atmospheric rivers (ARs) and events compounded by extreme precipitation and high coastal water levels throughout the Dutch coast. I feel that this article is very suited for the particular region and it presents a major innovation that should be considered for publication after a certain number of improvements (most of them minor).

I fully agree with the revision provided by Anonymous Referee #1, which I think is very complete and will help to improve the manuscript substantially. I just would add some comments:

**Major Comments:**

My main concern is the statistical strength of the results obtained here. I am not a close friend to complicate the analysis with statistical test when they are not really necessary, but in this case I think they are.

For example, in section 4.1, you say that almost 20% of days have AR detections. Then you say that 28% of days in Delfzijl does not show show AR-CE association, but you are analyzing the same day or within +-1, which may become the former 20% in a 60% of the days that are considered in the analysis. So you are claiming that CEs occur 72% of the time in coincidence with a something that exists, in general, up to 60% of the time... can the null hypothesis be rejected with this values? Personally, I doubt it... I strongly suggest to include suitable statistical test in the final version of the manuscript.

**Minor Comments:**

P2 L14-17 : Please, update this figures regarding the poleward transport of water vapor, and the width/length ratio in ARs with Guan and Waliser, (2015).

Guan, B., & Waliser, D. E. (2015). Detection of atmospheric rivers: Evaluation and application of an algorithm for global studies. Journal of Geophysical Research: Atmospheres, 120(24), 12514-12535.

P2 L25 : Please, consider to add a sentence on the source regions of moisture for Atlantic ARs. Take a look at https://www.earth-syst-dynam.net/7/371/2016/esd-7-371-2016.html.

P3 L7 : Replace "EOBS" by "E-OBS".

P3 L21 : Please, add something like "when synoptic forcing conditions are favorable" after "precipitation events".

P3 L23 : Add more information about the stations. To whom they belong?

P4 L4-7 : I think that it is completely unnecessary to describe ERA-In. Please, consider to replace this needless description by a citation.

P5 L23 : "processes" is written two times in the same sentence. Consider to find an alternative.

P6 L24 : replace "winter six months" by "extended winter".

P10 L6 : Do you mean Fig. 8a?

P10 L27 : Please, consider to rewrite the title of this subsection.

P12 L14 : "we provide vital information"... consider to replace "vital" by "important", or similar.

Table 1 : Include the period (1979-2015) in the caption.

Figure 4 : Include "ARCEs", "no ARCES", etc... in each box of the Figure.

Figure 5 : This figure is very complete and helps a lot to understand the results, but, please, simplify the legend and be consisted. For example, if I understood properly, the only difference between red and black lines is AR and no-AR detection. Then, why do you say "days" for the red line, and "t-series" for the black one? The same applies to the dots.

Figure 6 : This results refer to "Den Helder" only. Please, clarify somewhere in the caption.

Figure A1 : Please, rewrite the last sentence in the caption.

---

## Author Comment (AC1) · 8 Oct 2018

We would like to thank Referee #1 for her/his detailed comments and review of our manuscript. We believe that by addressing these comments we were able to significantly correct our analysis and thus improve the manuscript. Below we address each point raised by Referee #1 (*marked in blue, italic*) individually. Our responses to comments are shown in black. Text passages from the manuscript are included in red with text changes highlighted by underlining them and choosing **red, bold** font.

Best regards,
Nina Ridder, Hylke de Vries and Sybren Drijfhout

*This paper presents a novel analysis of the association between atmospheric rivers (ARs) and compound events (concurrent high precipitation and high sea water level) along the Dutch coast. The study represents a step further to understand the impacts of ARs beyond the traditional focus on precipitation alone, and may help extend the consideration of ARs in situational awareness and forecast of extreme events to regions where ARs have received relatively less attention in the science and/or applications community.*

*The analysis procedures are sound for the most part, but needs improvement/amendment as described in my specific comments below. A major missing component is a robust accounting of the statistical significance in the differences between CEs with and without ARs, and between ARs with and without CEs. In the only case where significance test is conducted (Figure 7), the test results do not seem to make physical sense (see specific comments below), which makes me worry about whether the significance test was properly conducted.*

We thank the referee for highlighting this shortcoming of the previous version of our manuscript. We revised our significance analysis used to produce Figure 7 and extended its application to the rest of the parameters as suggested by the referee. In detail, we now apply a student t-test that compares the anomalies (relative to monthly climatology) of the daily mean value of each variable during CEs (Fig. 6, 7 and 8) and ARs without CEs (Fig. 9) to the anomalies (relative to monthly climatology) of the daily mean value of each variable in the full time series. Further, we corrected the caption of Figure 7 to clarify that statistical significance is defined for areas with a p-values lower than or equal to (≤) 0.05.

*Specific comments:*

*Near Line 5: "accompanied by the presence of an AR", "up to seven days before": does this mean an event is considered AR-accompanied if an AR is*

*present up to seven days before the event? In any case, it would be useful to define "accompanied by an AR".*

This formulation, in the first version of the manuscript, might have been conveying a confusing message. We intended to express that we isolated the conditions before events with a lead-time of up to seven days. We rephrased the abstract as follows:

"[…] we find that the majority of compound events (CEs) between 1979 -2015 has been accompanied by the presence of an AR over the Netherlands**. In detail, we show that CEs have a three to four times higher chance of occurrence on days with an AR over the Netherlands compared to any random day (i.e. days without knowledge on presence of an AR). In contrast, the occurrence of a CE on a day without AR is three times less likely than on any random day. Additionally, by isolating and assessing the prevailing sea level pressure (SLP) and sea surface temperature (SST) conditions with and without AR involvement up to seven days before the events**, we show […]"

*Near line 10: "local ARs": it is not totally clear what "local" means here. Some ARs travel a longer distance than other ARs, but I'm sure that's what "local" aims to convey here.*

We intended to highlight that the AR has to occur over the study area to be able to influence the conditions during a compound event. We removed the word 'local' to prevent confusion and reformulated the sentence slightly to:

"These conditions are clearly distinguishable from those conditions during compound events without the influence of **an AR** which occur under SLP conditions resembling the East Atlantic (EA) pattern  […]"

*Near line 5: "sever": typo of "severe".*
Resolved.

*Near line 5: "future development of future flood risk": awkward construction.*
We changed this part to "[…] the future development of  flood risk."

*Near line 15: "in relation with extra-tropical cyclones": it would be more consistent with the definition in AMS Glossary of Meteorology to say "typically in relation with . . ."*
Added. The sentence now reads:
"They **typically** develop in relation with extra-tropical cyclones […]"

*Near line 15: "400 - 600 km": add a reference for the quantitative description, or make it qualitative with something like "several hundred km".*
Done.

*Near line 25: "a characteristic not previously assessed": change to something like ", a characteristic not previously assessed for ARs affecting the Europe", because there's at least one study that has examined the effect of ARs on sea water level in western US; see https://agupubs.onlinelibrary.wiley.com/doi/abstract/10.1002/2016GL070086*

We adjusted the manuscript to:

"a characteristic not previously assessed **for ARs affecting Europe** […]"

*Near line 30: "projected frequency enhancement and intensification of ARs": Espinoza et al. 2018 could also be cited here to support this statement where they systematically examined and compared such changes across the globe; https://agupubs.onlinelibrary.wiley.com/doi/10.1029/2017GL076968*

We added the mentioned reference to our citations and the relevant section in our manuscript.

*Near line 5: change "on both, " to "on both".*

Done.

*Near line 10: change "both, precipitation and water level," to "both precipitation and water level".*

Done.

*Near line 10: "identify days with the presence of an AR": for the sake of symmetry with CEs, a brief, high-level description of how AR days are identified is warranted here, i.e., based on certain quantile thresholds on intensity and geometry?*

We agree with the Reviewer that this statement needs more explanation. However, to keep the Introduction concise, we chose to explain this concept in the Methods Section (Sect. 3) of the manuscript instead. We added a reference to the description to the Introduction.

The last paragraph of the Method Section now reads as follows:

"**As mentioned in Section 2 the study presented in this paper isolates ARs in the database that passed over the Netherlands. For this, we isolated all days from the AR database on which an AR was detected within a box over 3.0˚E-7.2˚E/50.0˚N-54.0˚N (approximate location of the Netherlands) during at least one of the four daily time steps. This results in the equivalent treatment of days with an AR over the study area during multiple time steps and those days with an AR during only one time step. The duration of the presence of an AR over the study area is therefore neglected. This choice accounts for the frequency limitation set by the E-OBS dataset, which provides daily precipitation sums only (see Section 3.3).**"

*Near line 20: change "namely" to "namely,".*
Done.

*Near line 15: "and provided online by Bin Guan": consider removing as the information like this should be (and already is) in the acknowledgement section.*
We removed this part of the sentence.

*Near line 25: Guan et al. (2018) could also be cited here which provides more validation of the AR database based on comparing to field observations; see https://journals.ametsoc.org/doi/abs/10.1175/JHM-D-17-0114.1*
We added this reference to the relevant citation.

*Near line 15: "centred three-day precipitation": I have difficulty understanding what "centred" conveys in this sentence. That is, the word seems unnecessary. If precipitation amounts on day 1, 2, and 3 are a, b, and c mm, respectively, the 3-day precipitation is simply a+b+c mm, i.e., there's no "centering" needed to be done in the calculation.*
We apologise for this confusion. We clarified our definition as follows:

"[…] the  three-day precipitation sum over one of the chosen regions in the study area exceeds its 95th percentile and the total water level at the associated coastal station exceeds its 95th percentile at any point during the same three-day period. **The compound event is then considered to have occurred on the day in the centre of the three-day period over which the precipitation sum and the water level maximum was derived. The day before and after this are not considered compound events unless they are located in the middle of a three-day period that fulfils the above defined requirements.**"

*Near line 20: "number of compound events": the numbers are not fully meaningful without first defining what an "event" is, i.e., is an event counted as a day, a 3-day period, or a continuous period >=3 days?*
We hope that our adjustment mentioned in our response to the Reviewer's previous comment resolves this problem.

*Near line 20: "within +-1 days of the event": Now I sort of understand what "centred" meant in the earlier sentence. In the example I gave above, does it mean the resulting value of a+b+c is assigned to day 2, and the 3-day period centered on day-2 is considered AR-related if an AR occurred on one or more days of day 1, 2, or 3? Please use the answer to make clarifications in the data section in terms of how a CE is defined, how an "event" is counted (e.g., if a CE lasted 6 continuous days, is it counted as one event, 2 events, or 6 events?), when a CE is considered to be AR-related or not ARrelated, what*

*"day of event" means, etc. Without clear and unambiguous definitions of terms, the statistics presented are hard to make sense of.*

We apologise for this confusion. We clarified our definition by adding the following to the end of the section:

"[…] at any point during the same three-day period. **The compound event is then considered to have occurred on the day in the centre of the three-day period over which the precipitation sum and the water level maximum was derived. The day before and after this are not considered compound events unless they are located in the middle of a three-day period that fulfils the above defined requirements.**"

*Near line 30: "climatological" is a typo of "climatology", and "esembling" a typo of "resembling".*

Corrected.

*Near line 15: "probability density": for a probability density function, if the function is integrated over all possibilities, the result should be one. But that does not appear to be the case in Figure 5. If you integrate the values over the x-y plane in Figure 5, what does the resulting number represent? That determines how the values contoured in the figure should be called.*

We apologise for not providing a sufficient description of what Figure 5 is conveying. We added the following explanation to the caption of the Figure, which now is as follows:

"Joint probability distribution of three-day precipitation sums (mm) and three-day maximum total water level (m). **Contours denote the area enclosing indicated percentage of data (30, 50, 70, 90, 95 and 99\% contours are shown). Dark/red contours show data for days without/with an AR over the Netherlands.** Scatter plot […] "

*Near line 20: add "for" in front of "compound events without".*

Done.

*Near line 10: "absolut" is a typo of "absolute".*

Corrected.

*Near line 15: "persistent throughout the week before an event": this makes me think that there are conditions during the week prior to the AR that favors the development of warm SSTs and the AR, and in that regard the ARCE (AR+CE) perhaps should be emphasized as indicative of the interplay between these conditions, instead of one causing the other.*

We agree with the reviewer's comment and highlighted this in the manuscript by adding the following sentence to the relevant passage:

"The changes in SLP conditions are also reflected in the anomalies in sea surface temperature (Fig. 7) through the connection between surface winds and ocean currents. **This leads to spatial patterns that indicate the occurrence of compound events and provide a tool to predict the kind of compound event that will occur, i.e. CEs with AR association or CEs without.** In case of ARs with CEs […]"

*Near line 15: "loose" is a typo of "lose".*
Corrected.

*Near line 25: "Difference AR with CE and those without": please fix the grammar.*
We corrected the section title to
" Difference **between** AR**s** with **and without association to CEs**"

*Near line 5: "noARCEs": did you mean "noCEARs"? This makes think whether there's a better way to name these events that works better for both the authors and readers, because names like noARCEs and noCEARs are just a bit too cryptic, and when used together with names like ARCEs and CEARs (which I think are identical?) they may cause unnecessary confusions to both the authors and the readers. How about something more descriptive like the following: - CEs with ARs - CEs without ARs - ARs with CEs (identical to CEs with ARs) - ARs without CEs*
We agree with the Referee that the choice of abbreviation starts to be confusing in this section of the manuscript. We chose to replace the abbreviations noCEARs and CEARs with "ARs without CEs" and "ARs with CEs" respectively.

*Near line 30: "early identification of compound events . . . one week in advance": to make this statement and, more importantly, to make the main analysis of the paper more compelling, it is recommended to show that precursor conditions during the week leading to the CEs are statistically different than conditions leading to no CEs. It would be convenient to build on Figure 9 for this purpose, i.e., by expanding it to include the week before (similar to Figures 6 and 7), and adding significance test for the difference between "ARs with CEs" and "ARs without CEs". Significance test is also suggested to be added to Figures 6 and 8 and fixed in Figure 7 for the difference between "CEs with ARs" and "CEs without ARs". The paper heavily relies on statistical analysis (as opposed to dynamics-oriented analysis), so a robust accounting of the statistics is highly desirable.*
We added an analysis of the statistical significance of the shown SLP and precipitation anomalies as requested by the Referee and adjusted the relevant figures accordingly. We think with the additional analysis we delivered results that sufficiently support this statement.

We also added a significance test the anomalies during ARs without CEs and adjusted the text accordingly. We added a final paragraph of Section 4.4 to describe this:

"[…] **All features described above that characterise the mean conditions during ARs without CEs and make them different to the conditions during ARs with CEs are statistically significant (dotted areas in Fig. 6 – 9). This opens the possibility to use the here presented results in the early identification of an upcoming event."**

All adjusted Figures can be found at the end of this letter.

*Near line 10: "a specific definition of ARs": this sounds like there're many different definitions, which I don't think is true. My opinion is that the diversification in AR detection methods (perhaps 20 methods or more exist now) is a manifestation of the difficulty in detecting ARs, not because there're that many different definitions.*

We agree with the Referee that this formulation is misleading. We therefore changed this sentence to:

" We also note that the **identification of ARs that are analysed in this study is influenced by the applied AR-detection algorithm**. The particular algorithm applied here […]"

*Near line 15: "their effect would be marginal": consider removing this statement given the large variations across different AR detection methods (see https://www.geoscimodel-dev.net/11/2455/2018/).*

Removed.

*Near line 25: change "based on their poleward transport" to "based on their lacking of poleward transport".*

Done.

*Table 1 and where applicable in the text: "on day of event", "one day before or after event": given that the precipitation is a 3-day total, and CEs are defined using a 3-day window, descriptions like these are quite ambiguous. For example, if a CE occurred during the period of January 1-3, then common sense is that "one day before event" is December 31, and "one day after event" is "January 4". But that doesn't seem to be what the authors intended in indicate here. Again, an unambiguous definition of terms is needed to avoid potential confusions of this kind, as also suggested earlier.*

We hope that our adjustment mentioned in our response to the Reviewer's previous comment resolves this problem.

*Figure 2 caption: please define "area covered by AR", or how it was calculated. Area would have units of mˆ2, but it doesn't seem to be the case here. Did you mean AR frequency of occurrence (percent of time steps)? The latter is a more widely used and understood terminology in at least the AR community.*

We adjusted this caption to convey the information more clearly. The caption now reads:

"Climatology of daily mean sea level pressure (SLP; colour shading). **Contours mark regions over which ARs are located. Numbers indicate the relative amount of time that the respective area is covered by an ARs** throughout the study period (1979-2015)."

*Figure 3: "over NL": what does NL refer to or is it defined somewhere? Are the numbers per single month (i.e., the climatological mean), or the total over the given month? Suppose something happens 3 times in January, and is repeated for the past 100 years, it is more sensible to say it happens 3 times per month, instead of 300 times per month, right?*

We added the abbreviation "NL" to the figure caption as well as in the text within the Introduction and the Results section. Additionally we clarified that the numbers presented in the figure are monthly climatological mean values. The caption now reads:

"**Monthly climatological mean** number of compound events per month at the four coastal stations assessed in this study. Black columns indicate the number of all CEs (CEs with AR + CEs without AR + ARpm1dayCEs), while red bars show the number of CEs with association to an AR over the Netherlands (**NL**; CEs with AR)."

*Figure 6: the plots and fonts are too small. Also, the caption says "The right two columns" twice, the first one of which should be "The left two columns".*

We corrected the caption and increased the fontsizes in this Figure. The new figure can be found below.

*Figures 6, 7, 8: "Anomalies CE with AR" etc.: please change to "Anomalies during CEs with ARs", etc. for clarity.*

Done.

*Figure 7 caption: "Grey areas mark regions with a p-value below 0.05": given that a small p-value indicates high significance, do you mean the grey areas are where the values are significant, and the color shadings are where the values are NOT significant? That makes no sense because that would mean you are highlighting the nonsignificant values, and obscuring the significant values. Also, it is against intuition that the strongest anomaly values (darkest shading in the figure) are with large p-values, i.e., non-significant.*

We thank the Referee for bringing this problem to our attention. As mentioned above we have adjusted our statistical method to determine significance and corrected the Figure and caption (see end of this letter). We also changed the text passages referring to this figure.

"[…] As a result, the wind anomalies, which increase with time getting stronger closer to the event (Fig. A2), induce a **decrease in SSTs within the North Atlantic subpolar gyre that expands throughout the week before the event (Fig. 7a, c and e). On the day of the event this negative anomaly covers parts of the Labrador Sea and the subpolar North Atlantic. At the same time an** increase in SSTs develops that covers large parts of the western and central (tropical and subtropical) North Atlantic, the North Sea and parts of the Norwegian Sea on the day of the event (Fig. 7a, c and e). […] **The negative SST anomaly pattern over the subpolar North Atlantic is most likely caused by changes in the transport of surface waters from higher latitudes to subpolar North Atlantic due to a strengthening of the north-northeasterly component of the wind field throughout the week before the event (Fig. A2).** However, […]"

**Figures**

[Figure]

Figure 6: Temporal evolution of mean conditions seven (a-d) and four days (e-h) before a CE at Den Helder and on the day of the event itself (i-l). The left two columns, i.e. panels a, e, i and b, f, j, show the evolution of anomalies in SLP (colour shading) and IVT (vector field) during CEs with and without AR association, respectively. The right two columns, i.e. panels c, g, k and d, h, l, show the same but for absolute values of daily mean SLP and IVT. Results for the three other stations (not shown) are comparable. **Stippled areas mark regions with a p-value below 0.05 derived from student t-test of daily mean SLP values compared to the daily mean values of full time series.**

[Figure]

Figure 7: Anomalies in daily mean SSTs (shading) related to CEs with (left panels a, c and e) and CEs without AR association (right panels b, d and f) seven and four days before a CE (a and b; c and d, respectively) and on the day of the event (e and f). Contours mark regions that are occupied by more than 30% of all ARs in the specific category with contour intervals at 30%, 40%, 60%, 80%, 90%, 99% and 100%. **Stippled areas mark regions with a p-value below 0.05 derived from a student t-test comparing the monthly anomalies of daily mean SST values on the day of events to those throughout the full time series."**

[Figure]

Figure 8: Anomalies of daily mean precipitation sums during CEs with (a) and without (b) AR association. **Stippled areas mark regions with a p-value below 0.05 derived from a student t-test of daily precipitation values during events and the full time series**.

[Figure]

Figure 9: Anomalies of (a) SLP (colour shading) and IVT (vectors), (b) SST (colour shading) and relative number of ARs covering an area, and (c) precipitation on days with an AR over the Netherlands without the occurrence

of a CE. **Stippled areas indicate regions where the difference in conditions between ARs with CEs and ARs without CEs are statistical significant with a p-value below 0.05 derived from a student t-test comparing monthly anomalies of daily mean values during events to those of the full time series.**

---

## Author Comment (AC2) · 8 Oct 2018

We would like to thank Referee #2 for her/his comments. Addressing these comments has helped to improve the manuscript. Below we address each point raised by Referee #2 (*marked in blue, italic*) individually. Our responses to comments are shown in black. Text passages from the manuscript are included in red with text changes highlighted by underlining them and choosing **red, bold** font.

Best regards,
Nina Ridder, Hylke de Vries and Sybren Drijfhout

*My main concern is the statistical strength of the results obtained here. I am not a close friend to complicate the analysis with statistical test when they are not really necessary, but in this case I think they are. For example, in section 4.1, you say that almost 20% of days have AR detections. Then you say that 28% of days in Delfzijl does not show show AR-CE association, but you are analyzing the same day or within +-1, which may become the former 20% in a 60% of the days that are considered in the analysis. So you are claiming that CEs occur 72% of the time in coincidence with a something that exists, in general, up to 60% of the time… can the null hypothesis be rejected with this values? Personally, I doubt it… I strongly suggest to include suitable statistical test in the final version of the manuscript.*

If we understand the referee's concerns correctly, the referee is concerned about the significance of our finding in regards to the association of CEs to ARs compared to climatology. The Referee argues that due to the 20% chance of an AR at any given day, the chance of an AR occurring over a three-day period should be 60%. We do not see this in our data. We performed a simple test by counting the days without the presence of an AR over the Netherlands over a three-day period and compared it to the total number of days in the study period. We find that these days make up 61% of the total days, while those days with an AR over a three-day period occur only 39% of the time. Thus, the ratio of days without ARs (including +/-1day) and days with ARs on the day or the day before or after between climatology and CEs is significantly different, i.e. 61:39 (climatology) vs. 28:72 (CEs at Delfzijl). We therefore think that our conclusion that ARs play an important role in the occurrence of CEs is sufficiently supported by our results. To underline this in the manuscript we added the following sentences at the end of the first paragraph of Section 4.1:

"Only a small fraction of 18% (Harlingen) to 28% (Delfzijl) of CEs does not show any association to the presence of an AR over the Netherlands. **This is significantly different to climatology with roughly 61% of days that lack the presence of an AR over a three-day period against 39% of days with an AR detected over the Netherlands either on the day itself or the day before or after. As a result, the chance of having a CE on a random day (ie. without knowledge on presence AR) is a factor three higher than**

**that on a day without an AR, whereas the chance on having a CE on a day with AR is a factor three to four higher than on a random day."**

*Minor Comments:*
*P2 L14-17 : Please, update this figures regarding the poleward transport of water vapor, and the width/length ratio in ARs with Guan and Waliser, (2015). Guan, B., & Waliser, D. E. (2015). Detection of atmospheric rivers: Evaluation and application of an algorithm for global studies. Journal of Geophysical Research: Atmospheres, 120(24), 12514-12535.*
We adjusted the manuscript as follows:
The vast geometric extent of ARs with a typical width of **several hundred kilometres (<1,000 km)** and lengths of over 2,000 km allows them to cover and affect large geographical areas simultaneously (Ralph et al. 2004; Guan and Waliser, 2015).

*P2 L25 : Please, consider to add a sentence on the source regions of moisture for Atlantic ARs. Take a look at https://www.earth-syst-dynam.net/7/371/2016/esd-7-371-2016.html.*
We added the following sentence to address this:
"[…] They [ARs] typically develop in relation with extra-tropical cyclones and move with the large-scale dynamic phenomena that produce them (hereafter AR system). **In the case of Western Europe, the moisture contained in ARs hitting this region originates from evaporation over an area stretching from the subtropical North Atlantic (north of 20˚N) over the central and western North Atlantic to the West European coast (Ramos et al., 2016).** The vast geometric […]"

*P3 L7 : Replace "EOBS" by "E-OBS".*
Done.

*P3 L21 : Please, add something like "when synoptic forcing conditions are favorable" after "precipitation events".*
We added the following:
Nevertheless, it has been shown that those ARs making landfall along the Dutch coast can lead to significant precipitation events **depending on the forcing conditions caused by the prevailing large-scale atmospheric conditions** (Waliser and Guan, 2017).

*P3 L23 : Add more information about the stations. To whom they belong?*
This section now reads:
[…] and the north-east of the Netherlands (hereafter NENL) for Delfzijl. **All stations are operated by Dutch Ministry of Infrastrcture and Water Management and are located in four different water boards**. The stations

were chosen […]

*P4 L4-7 : I think that it is completely unnecessary to describe ERA-In. Please, consider to replace this needless description by a citation.*
We followed the reviewer's advice and deleted part of the description. The section now reads:
The ERA-Interim reanalysis dataset is produced by the European Centre for Medium-Range Weather Forecast (ECMWF). It is the result of reanalysis simulations performed using a three-component forecast model (Integrated Forecasting System IFS release Cy31r2) for the time period from 1 Jan 1979 to present day (Berrisford et al., 2011; Dee et al., 2011).  This study uses data for mean sea level pressure, zonal and meridional wind components to force a numerical storm surge model, […]

*P5 L23 : "processes" is written two times in the same sentence. Consider to find an alternative.*
We replaced the second "processes" with the word "mechanisms".

*P6 L24 : replace "winter six months" by "extended winter".*
Done.

*P10 L6 : Do you mean Fig. 8a?*
Reference was adjusted from 8b to 8a.

*P10 L27 : Please, consider to rewrite the title of this subsection.*
The new section title now reads:
**Difference between ARs with and without association to CEs**

*P12 L14 : "we provide vital information"… consider to replace "vital" by "important", or similar.*
We changed "vital" to "crucial"

*Table 1 : Include the period (1979-2015) in the caption.*
Done.

*Figure 4 : Include "ARCEs", "no ARCES", etc… in each box of the Figure.*
We added the labels requested by the reviewer to the Figure (see below).

[Figure]

*Figure 5 : This figure is very complete and helps a lot to understand the results, but, please, simplify the legend and be consisted. For example, if I understood properly, the only difference between red and black lines is AR and no-AR detection. Then, why do you say "days" for the red line, and "t-series" for the black one? The same applies to the dots.*

We adjusted the caption and legend of the figure according to the Referee's suggestion (see below).

[Figure]

Legend:
- days without AR over NL (black line)
- days with AR over NL (red line)
- days without AR over NL (grey dots)
- days with AR over NL (magenta dots)
- median days without AR over NL (black X)
- median days with AR over NL (red X)
- median CEs without AR over NL (black +)
- median CEs with AR over NL (red +)

*Figure 6 : This results refer to "Den Helder" only. Please, clarify somewhere in the caption.*

We adjusted the caption of this figure to the following:

Temporal evolution of mean conditions seven (a-d) and four days (e-h) before a CE **at Den Helder** and on the day of the event itself (i-l). The left two columns, i.e. panels a, e, i and b, f, j, show the evolution of anomalies in SLP

(colour shading) and IVT (vector field) during CEs with and without AR association, respectively. The right two columns, i.e. panels c, g, k and d, h, l, show the same but for absolute values of daily mean SLP and IVT. **Results for the three other stations (not shown) are comparable.**

*Figure A1 : Please, rewrite the last sentence in the caption.*
The last sentence of the caption now reads:
**Dashed horizontal lines indicate the climatological values for the different landfall locations.**

---

## Author Response (AR2)

We thank the reviewer for his/her helpful comments, which in our opinion helped to improve the clarity of the manuscript. Below our detailed response (black) to the comments raised by the reviewer (blue) including changes made in the manuscript (red).

Best wishes,
Nina Ridder, Hylke de Vries and Sybren Drijfhout

1. Near line 20: "number of compound events": the numbers are not fully meaningful without first defining what an "event" is, i.e., is an event counted as a day, a 3-day period, or a continuous period >=3 days? (Note: Examples will always help. Suppose the event thresholds are met for all days from January 1st to 3rd, i.e., each of the 3-day window centered on Jan 1, Jan 2, and Jan 3 satisfies the event threshold), is this period counted as one event, or 3 events?)

An event is considered to be one day only. To illustrate this on an example and to address the second comment of the reviewer we included the following explanation to the end of Sect. 3.5 - "Definition of compound events":

"*The compound event is then considered to have occurred on the day in the centre of the three-day period over which the precipitation sum and the water level maximum was derived. The day before and after this are not considered compound events unless they are located in the middle of a three-day period that fulfills the above defined requirements.* For instance, suppose that the precipitation (P) summed over day_1, day_2 and day_3 exceeds the 95$^{th}$ percentile of the three-day precipitation sum of the full time series (p $_{95\%}^{P}$ ), i.e. $\sum_{n=1}^{3} P(day\_n) > p_{95}^{P}$ . If the maximum total water level ( $H_{max}^{d1,d2,d3}$ ) during the same three-day period exceeds the 95th percentile of the total water level (H) of the full time series ( $p_{95\%}^{H}$ ), i.e. $H_{max}^{d1,d2,d3} = max(H(day\_1), H(day\_2), H(day\_3)) > p_{95\%}^{H}$ , then day_2 is considered to be a "compound event". Day_1 and day_3 are not defined as compound events, unless they themselves fulfill the definition of compound events, i.e. for day_1: $\sum_{n=0}^{2} P(day\_n) > p_{95}^{P} \wedge max(H(day\_0), H(day\_1), H(day\_2)) > p_{95\%}^{H}$ and for day_3: $\sum_{n=2}^{4} P(day\_n) > p_{95}^{P} \wedge max(H(day\_2), H(day\_3), H(day\_4)) > p_{95\%}^{H}$ .
Each of the three days is then counted as a single compound event. As a result, should all three days (day_1, day_2 and day_3) fulfill the requirement of compound events then they are considered as three separate events. Accordingly, the day before the compound event on day_1 would be day_0, the day before CE on day_2 would be day_1 etc."

2. Near line 20: "within +-1 days of the event": Now I sort of understand what "centred" meant in the earlier sentence. In the example I gave above, does it mean the

resulting value of a+b+c is assigned to day 2, and the 3-day period centered on day-2 is considered AR-related if an AR occurred on one or more days of day 1, 2, or 3? Please use the answer to make clarifications in the data section in terms of how a CE is defined, how an "event" is counted (e.g., if a CE lasted 6 continuous days, is it counted as one event, 2 events, or 6 events?), when a CE is considered to be AR-related or not AR-related, what "day of event" means, etc. Without clear and unambiguous definitions of terms, the statistics presented are hard to make sense of. (Note: Please directly address each point in the above comment if possible, and use examples where they might help the explanation. Among these, the two key questions are how the events are counted, and when a CE (defined using a 3-day window but assigned to the middle day) and an AR (which presumably is based on 6-hourly time steps) are considered to be associated with each other. I understand some of the answers would seem too obvious to the authors, but the hope here is that the description should be unambiguous enough for an ordinary reader to understand the statistics and/or re-produce the analysis procedures.)

Please refer to our answer to the reviewer's previous comment.

3. Table 1 and where applicable in the text: "on day of event", "one day before or after event": given that the precipitation is a 3-day total, and CEs are defined using a 3-day window, descriptions like these are quite ambiguous. For example, if a CE occurred during the period of January 1-3, then common sense is that "one day before event" is December 31, and "one day after event" is "January 4". But that doesn't seem to be what the authors intended in indicate here. Again, an unambiguous definition of terms is needed to avoid potential confusions of this kind, as also suggested earlier. (Note: The authors did address this comment. But now that I understand better how an event is defined, a new question arises: how is "one day before event" and "one day after event" dealt with in cases where multiple events occur consecutively? Let's use the earlier example again, i.e., assume the event thresholds are met for all days from January 1st to 3rd, i.e., each of the 3-day window centered on Jan 1, Jan 2, and Jan 3 satisfies the event threshold. Then, from the perspective of the Jan 2 event, Jan 1 and Jan 3 would be considered "one day before event" and "one day after event". But that leads to a paradox, because both Jan 1 and Jan 3 are themselves "day of event". Similarly, although Jan 2 is a "day of event", it is also "one day before event" because it precedes the Jan 3 event, and is also "one day after event" because it follows the Jan 1 event. In other words, all of your statistics for "day of event", "day before event", and "day after event" are, inadvertently, a mixture of "day of event", "day before event", and "day after event". This issue complicates the physical interpretation of the statistics and should be remedied/mitigated.)

The reviewer raises a valid point with this concern and highlights that the manuscript wasn't clear enough in the explanation of the used terminology. When referring to "CEs

with an AR over the Netherlands", we intended to refer to the set of compound events that coincides with the presence of an AR only on the day of the event itself. There is no AR present on the day before or after the event. Hence, if we take the example from our previous answer, if day_2 is considered to be a CE and classified as a 'CE with an AR over the Netherlands', neither day_1 nor day_3 have an AR over the Netherlands at any of their six timesteps. Therefore, our statistical analysis is limited to the comparison of (i) CEs with ARs over the Netherlands (on the same day as the event itself and without ARs on the day before and/or after) and (ii) CEs without an AR over the Netherlands on the day of the event (regardless of the presence of an AR on the days before and/or after the day of the CE). Neither of the two sets includes CEs where an AR is present on the day before or after the event day. We included a clarification of this into the manuscript.

In the second paragraph of Sect. 4.1 - "Climatology":
"... *To allow a thorough investigation of the impact of ARs on CEs in this study, we differentiate three types of CEs, namely those events that co-occur with an AR over the Netherlands, either on the day of the event (hereafter CEs with AR; Fig. 4b) or one day before and/or after (hereafter CEs with AR ± 1day; Fig. 4c), and those that occur in the absence of an AR in the three days around the event (hereafter CEs without AR; Fig. 4d).* To illustrate the difference between these types of event, remember the example used in Sect. 3.5. Assume day_2 is a CE. If no AR is detected at any of the timesteps throughout day_1, day_2 and day_3, day_2 is considered a "CE without AR". Should there be no ARs over the Netherlands on day_2 but on either day_1 and/or day_3, i.e. the day before and/or after the CE, then day_2 is considered to be a "CE with AR ± 1day". Only if there is an AR over the Netherlands during one of the timesteps of day_2 itself, day_2 is considered a "CE with AR". *Since the latter two types of CEs, i.e. CEs with AR and CEs with AR ± 1day, show very similar atmospheric climatological anomalies …*"

At the end of the caption of Table 1:

"... For an explanation of terminologies and a definition of the different types of compound events please refer to the text."